# Assessing the Fragility of SHAP-Based Model Explanations Using Counterfactuals

Cornelia C. Käsbohrer[*1], Sebastian Mair[2], and Lili Jiang[1]

[1]Umeå University, Sweden
[2]Linköping University, Sweden
cornelia.kaesbohrer@cs.umu.se

## Abstract

Post-hoc explanations such as SHAP are increasingly used to justify machine learning predictions. Yet, these explanations can be fragile: small, realistic input perturbations can cause large shifts in the importance of attributed features. We present a multi-seed, distance-controlled *stability assessment* for SHAP-based model explanations. For each data instance, we use DiCE to generate plausible counterfactuals, pool across random seeds, deduplicate, and retain the $K$ nearest counterfactuals. Using a shared independent masker and the model's logit (raw margin), we measure per-feature attribution shifts and summarise instance-level instability. On four tabular fairness benchmark datasets, we apply our protocol to a logistic regression, a multilayer perceptron, and decision trees, including boosted and bagged versions. We report within-model group-wise explanation stability and examine which features most often drive the observed shifts. To contextualise our findings, we additionally report coverage, effective-$K$, distance-to-boundary, and outlier diagnostics. The protocol is model-agnostic yet practical for deep networks (batched inference, shared background), turning explanation variability into an actionable fairness assessment without altering trained models.

## 1 Introduction

Explainable AI (xAI) approaches such as post-hoc explanations using SHAP [1] are increasingly being paired with deep learning systems to justify predictions in high-stakes settings, e.g., science, hiring, justice, or healthcare [2–4]. Stakeholders use these "reasons" to contest outcomes, debug models, and satisfy governance requirements. Yet a persistent practical problem remains: even small, feasible perturbations that cross the decision boundary can lead to considerably different attributions. We therefore pose a process-level question: are local explanations of a model *stable* under small, feasible edits that flip the decision, and equivalently, do similar individuals receive consistent reasons? We introduce a stability audit to evaluate this empirically. Rather than making normative fairness claims, we treat instability as a diagnostic of process consistency that can inform procedural assessments.

Prior work has demonstrated that counterfactual explanation generators can be sensitive to initialization and search configurations [5], and that post-hoc explanations can display group-based disparities in quality across different subpopulations [6]. Robustness studies typically probe explanations either by perturbing inputs with largely isotropic noise, which is not tailored to decision-relevant directions, or by relying on a particular counterfactual (CF) solver, so that solver randomness becomes entangled with any model-level instability being measured [7–9]. Fairness audits, in turn, largely focus on outcome disparities rather than on whether different groups receive explanations that are equally stable [10, 11]. What is missing is a model-class-agnostic, multi-seed, distance-controlled protocol that uses decision-relevant perturbations to measure explanation stability at scale, including for deep neural networks.

We propose an assessment of SHAP-based [1] model explanation stability under *plausible edits*. A *counterfactual* for an instance is a nearby, feasible version of the same case that flips the model's prediction, i.e., crosses the decision boundary. We generate CF candidates with Diverse Counterfactual Explanations (DiCE) [12] across multiple random seeds, pool and deduplicate them, and then retain only the closest $K$ decision-flipping edits to enforce a controlled neighbourhood around each instance.

Using a shared independent masker and explaining the model's margin/logit, we compute SHAP attributions for the instance and for each retained CF. For each feature, we calculate the median absolute change across the CFs and aggregate these changes into an instability score, $I_2$ (root-sum-of-squares). This allows us to contextualise the results and distinguish generator anomalies from model behaviour. We also report on lightweight diagnostics including coverage (the proportion of instances with at least one CF), effective $K$ (the number of CFs that were actually retained), proximity to the decision boundary (measured via the absolute margin), diversity among a person's CFs and a simple outlier rate using the Local Outlier Factor (LOF) [13].

---

*Corresponding Author.

Proceedings of the 7th Northern Lights Deep Learning Conference (NLDL), PMLR 307, 2026.

Empirically, across common fairness benchmarks and multiple architectures (i.e., logistic regression (LR), tree ensembles, and multi-layer perceptrons (MLPs)), we find consistent patterns. We can observe that instability increases with distance to the decision boundary. Tree models exhibit higher typical and tail instability than LR/MLP. After conditioning on proximity, most dataset-model pairs show overlapping group distributions, though a few display small but systematic protected-reference gaps, hence procedural-fairness risk signals under our setup. High coverage and effective-$K$ close to the target indicate these findings are not driven by CF search anomalies.

Our contributions are:

1. a *multi-seed, distance-controlled* CF neighbourhood that mitigates solver stochasticity,

2. a SHAP-based model-agnostic stability score with practical diagnostics,

3. an empirical study showing when instability concentrates (by proximity, model class) and when groupwise risk signals persist after proximity control.

The assessment is diagnostic (not normative) and slots alongside accuracy and outcome-based fairness checks to ask a complementary question: do similar individuals receive *stable reasons*?

## 2   Related Work

**Why explainability matters for neural networks.** Deep neural networks achieve often state-of-the-art accuracy but are opaque. Their capacity and nonlinearity make internal decision logic hard to inspect. In high-stakes settings, practitioners therefore rely on post-hoc explanations (e.g., SHAP) to debug, monitor, and justify predictions to stakeholders [14–17]. Because explanations shape human trust and downstream actions, disparities in *explanation quality* across groups can yield unequal outcomes even when the classifier is fixed. Dai et al. [6] formalise this concern along four axes—fidelity, stability, consistency, and sparsity—and document group disparities across datasets, models, and methods.

**Explanation stability and vulnerabilities.** Stability has emerged as a central quality criterion for xAI, with dedicated measures and taxonomies [18, 19]. Unstable or manipulable explanations can mislead users and "fairwash" models [20, 21]. Critiques also highlight regimes where SHAP is unreliable without careful protocol choices (e.g., link function, background data) [22]. Bridging explainability and fairness has been an active theme [17, 23, 24], with cautions about benchmark usage and data ethics [25–27].

**Counterfactuals as plausible variation and our positioning.** Counterfactual explanations operationalise feasible edits to inputs in black-box settings [12, 28–30], with work on efficiency/metrics [31] and fairness when used for recourse [32]. We adopt counterfactuals not to prescribe recourse but rather as a *probe of plausible and feasible variations*. This replaces random noise with feasible edits and allows us to explicitly control distance and seed effects to evaluate stability. Related efforts connect game-theoretic attributions and counterfactuals [33–38].

Compared to disparity analyses that perturb inputs using noise, our contribution is a *multi-seed, distance-controlled* counterfactual stability assessment. Specifically, we (i) pool and deduplicate DiCE proposals across seeds; (ii) select the $K$ nearest CFs in standardised space; (iii) compute SHAP shifts using a shared independent masker on the model's logit to ensure a consistent attribution protocol across models. This reframes stability as behaviour under *plausible edits* rather than random perturbations, while remaining model-agnostic and practical.

## 3   Stability Assessment Protocol

We formalise a model-agnostic procedure to quantify the stability of SHAP explanations under plausible and feasible (within DiCE) variations. This section defines the notation, attribution protocol, multi-seed counterfactual set, stability summaries, and fairness-oriented diagnostics that are used throughout the paper.

**Preliminaries and notation.** Let $\mathcal{D} = \{(x_i, s_i, y_i)\}_{i=1}^n$ be a tabular dataset with feature vectors $x_i \in \mathbb{R}^d$, a sensitive attribute $s_i$ (e.g., gender, race)[1], and binary labels $y_i \in \{0, 1\}$. Let $g_i \in \{0, 1\}$ index protected groups derived from $s_i$. A trained classifier $f : \mathbb{R}^d \to [0, 1]$ outputs $p(x) = \mathbb{P}(Y=1 \mid x)$. We explain the model's *raw margin* (i.e., the *logit*)

$$\ell(x) = \log \frac{p(x)}{1 - p(x)},$$

or the model's decision function when available. Explaining the margin avoids probability-scale compression and yields cleaner SHAP additivity. Model-specific details are deferred to Section 4.

**Background on SHAP.** SHAP [1] attributes a scalar prediction to features via Shapley values. We write $\phi \equiv \phi_{f,\mathcal{B}}^{\text{int}} : \mathbb{R}^d \to \mathbb{R}^d$ for the *permutation* SHAP explainer, which returns per-feature attributions for input $x$. With an independent masker and a fixed

---

[1]Note that $s_i$ is part of $x_i$, but the information is clearly indicated and is thus separately listed as $s_i$.

background $\mathcal{B}$ (a small subsample of the training data reused across models), coordinates outside a retained subset $S$ of all features are independently imputed from $Z \sim \mathcal{B}$. Here, $\phi_j(x)$ denotes the Shapley-weighted marginal contribution of feature $j$ averaged over coalitions $S$ and draws $Z$. We estimate these interventional values with *permutation SHAP* and always explain the margin/logit (LR decision function; MLP pre-sigmoid; $\mathrm{logit}(\hat{p})$ for trees), keeping the protocol identical across model classes. For context, *observational/conditional* SHAP conditions on the joint data distribution [39–41].

Across all model families (logistic regression, trees, MLPs) we use *permutation SHAP* with an independent masker and a single background $\mathcal{B}$ (fixed per dataset), and we always explain the margin/logit (LR decision function; MLP pre-sigmoid; $\mathrm{logit}(\hat{p})$ for trees). This yields interventional SHAP estimates under the independence assumption and keeps the protocol identical across model classes (we deliberately avoid TreeSHAP/DeepSHAP).

**Why SHAP (vs. LIME/Anchors) for stability assessment?** We measure per-feature shifts in attributions under small, plausible edits. SHAP provides local accuracy (additivity) and consistency [1], ensuring that changes in attribution aggregate meaningfully on the margin or logit. It also supports a protocol that remains fixed across model families by using an independent masker with a shared background $\mathcal{B}$ [39, 41]. LIME [42] fits a locally weighted surrogate whose coefficients depend on kernel width, neighbourhood sampling, and discretisation choices, which can inject method-induced variance. Note that LIME and SHAP can also be manipulated [20, 21]. Anchors provide rules with precision guarantees [43] but not vector attributions, making per-feature shift metrics ill-posed. Thus, we utilise SHAP and keep analyses within model, checking whether patterns replicate across models rather than comparing raw magnitudes.

**Counterfactual explanation via DiCE.** Counterfactual explanations ask how a model's prediction would change if certain features were altered. DiCE [12] proposes nearby points $x'$ that flip the decision, while supporting feasibility, sparsity, and actionability constraints. Unlike purely causal approaches that require a structural model [44, 45], DiCE operates directly on data and the trained predictor. CFs need not be single-feature interventions, so correlated features may change jointly. We therefore interpret SHAP shifts as *explanation sensitivity under plausible, identity-preserving edits*, rather than as causal effects. For each query instance $x$, we generate CF candidates under multiple seeds, pool, and deduplicate them, and retain the $K$ nearest CFs in a standardised feature space.

**Why DiCE (vs. other CF generators).** Our protocol needs a CF generator that is model-agnostic across logistic regression (LR), decision trees, and multi-layer perceptrons (MLPs), practical on tabular data without auxiliary generative models, compatible with distance control and seed pooling, and able to encode simple feasibility constraints. DiCE [12] satisfies these via random/gradient search with distance-based objectives. Alternatives trade off properties we do not require here: optimization-only CFs are flexible but sensitive to hyperparameters and slower on deep models [28]. Graph-based FACE assumes a reliable manifold and can be computationally heavier [29]. The contrastive explanations method [46] introduces autoencoders and model-specific tuning. Thus, we use DiCE as a probe of plausible variation and mitigate seed sensitivity via pooled, deduplicated, $K$-nearest candidates.

**Attribution protocol.** We explain the model's margin/logit $\ell(x)$ (decision function when available) rather than probability to avoid probability-scale compression and to preserve additivity. Throughout we use interventional SHAP with an independent masker: features outside a coalition are sampled independently from a fixed background $\mathcal{B}$ drawn from the training split. For each dataset, the same background $\mathcal{B}$ is shared across all models to ensure a consistent attribution protocol. All stability analyses are conducted within each model family and we additionally report whether patterns replicate across models. Implementation details are in Section 4.

**Multi-seed, distance-controlled counterfactual set.** For every query instance $x$, and for seeds $r = 1, \ldots, R$, let $\mathcal{C}_r(x)$ be the set of DiCE candidates that flip the decision. We pool candidates across seeds and deduplicate them afterwards, i.e.,

$$\mathcal{C}(x) = \mathsf{dedup}\left( \bigcup_{r=1}^{R} \mathcal{C}_r(x) \right).$$

Let $T$ be the per-feature standardisation map (continuous $j$: $T_j(u) = (u_j - \mu_j)/\sigma_j$ using train mean $\mu_j$ and std $\sigma_j$; one-hot categoricals remain unchanged), and define the distance

$$D(u, v) = \left\| T(u) - T(v) \right\|_2.$$

We then select the $K$ counterfactuals in $\mathcal{C}(x)$ that are closest to the query $x$ according to $D(\cdot, \cdot)$:

$$\mathcal{S}_K(x) = \mathrm{KNN}_K(x; \mathcal{C}(x), D).$$

The size of this set is denoted as $K_{\mathrm{eff}}(x) = |\mathcal{S}_K(x)|$ and we report the coverage as the fraction of instances with at least one selected CF $K_{\mathrm{eff}}(x) > 0$:

$$\mathrm{coverage} = \frac{1}{|\mathcal{X}|} \sum_{x \in \mathcal{X}} \mathbf{1}\{K_{\mathrm{eff}}(x) > 0\}.$$

Note that $K_{\text{eff}}(x) = \min\{K, |\mathcal{C}(x)|\} \leq K$ as the generator can yield less CFs. This *multi-seed (seed-pooled), distance-controlled* selection mitigates generator stochasticity while focusing on comparable, nearby edits. Conditioned on the fixed set $\mathcal{S}_K(x)$ and a fixed SHAP background $\mathcal{B}$, the subsequent calculation of attribution-shift is deterministic. As the number of seeds $R$ increases, dependence on any single seed correspondingly diminishes. This mitigates known sources of instabilities and the susceptibility of hill-climbing CF methods to manipulation [5].

**Explanation-shift measures.** Let $\phi(x) \in \mathbb{R}^d$ denote the SHAP vector for $\ell(x)$. For any counterfactual $z \in \mathcal{S}_K(x)$, define per-feature shifts

$$\Delta_j(x, z) = \bigl|\phi_j(x) - \phi_j(z)\bigr|, \qquad j = 1, \ldots, d.$$

We summarise the typical shift per feature by the median across the selected counterfactuals,

$$\widetilde{\Delta}_j(x) = \operatorname{median}_{z \in \mathcal{S}_K(x)} \Delta_j(x, z)$$

and aggregate to an instance-level instability via

$$I_2(x) = \bigl\|\widetilde{\Delta}(x)\bigr\|_2.$$

In summary, $I_2$ measures the total typical movement of attributions emphasising "spikes"—a few large shifts—due to its quadratic weighting. Units coincide with SHAP values on the logit scale, so comparisons of $I_2$ are made within each model/dataset, rather than across them. Lower values correspond to more stable explanations under plausible counterfactual perturbations, while higher values indicate fragility.

**Population and group summaries.** Let $\mathcal{X} = \{x_i\}_{i=1}^{\tilde{n}}$ be the evaluation set of query instances, a subset of the test inputs. For $g \in \{1, 2\}$ we define $\mathcal{X}_g = \{x_i \mid g_i = g\}$, where $g_i$ is the group label derived from the sensitive attributes. We report the distribution of $I_2(x_i)$ over $\mathcal{X}$ and compute feature-wise averages

$$\bar{\Delta}_j = \frac{1}{|\mathcal{X}|} \sum_{x \in \mathcal{X}} \widetilde{\Delta}_j(x), \qquad j = 1, \ldots, d.$$

For group $g$, the feature-wise average is

$$\bar{\Delta}_j^{(g)} = \frac{1}{|\mathcal{X}_g|} \sum_{x \in \mathcal{X}_g} \widetilde{\Delta}_j(x).$$

Within each model, we compare groups using robust summaries. We report the group medians of $I_2$ and their difference $\Delta_{\text{med}} = \operatorname{median}\{I_2(x) \mid g{=}1\} - \operatorname{median}\{I_2(x) \mid g{=}0\}$. Alongside the medians we show the interquartile range (IQR), defined as $Q_3 - Q_1$, as a robust measure of spread. We also report Cliff's $\delta$, a nonparametric effect

size that quantifies stochastic dominance between groups. Let $A$ denote the instability scores for group $g = 1$ and $B$ for group $g = 0$. Cliff's $\delta$ is given by $\delta = \Pr(A > B) - \Pr(A < B)$ which ranges over $[-1, 1]$, with $\delta = 0$ indicating no stochastic dominance. By convention $|\delta| \in [0.147, 0.33)$ is considered a small effect, $[0.33, 0.474)$ a medium effect, and $\geq 0.474$ a large effect. All reported differences and effect sizes include $95\%$ bootstrap confidence intervals. We compute these by resampling instances with replacement within each group (and within proximity bins when applicable) for $1,000$ replicates and taking the corresponding percentile interval. Because $\mathcal{S}_K(x)$ is multi-seed and distance-controlled, the resulting disparities are less confounded by counterfactual search variance and more indicative of genuine model behaviour.

**Proximity and equal-width binning for diagnostics.** To diagnose how instability varies with decision difficulty and to enable a fair comparisons between groups under the same conditions—so that the differences aren't due to confounders—we require a simple notion of proximity to the decision boundary and a robust way to condition on it. For each evaluation point $x$ we define a *proximity* score $r(x) = |\ell(x)|$ if the model exposes a margin/logit $\ell(x)$ or $\operatorname{median}_{z \in \mathcal{S}_K(x)} \|z - x\|_2$ otherwise as a CF distance proxy. The larger $r(x)$ is, the farther $x$ is from the decision boundary. We partition the range $[r_{\min}, r_{\max}]$ into $m$ bins with *equal width* and edges

$$a_k = r_{\min} + \frac{k}{m} (r_{\max} - r_{\min}), \qquad k = 0, \ldots, m.$$

We then assign each point to a bin $b(x) = \min\{k-1 : r(x) \leq a_k\}$. Within each bin, we summarise instability using the median and IQR of $I_2(x)$, and by analysing groups we determine the per-group medians and their gap. Binning provides us a nonparametric control for boundary proximity, avoiding functional-form assumptions. It enables "apples-to-apples" group comparisons at the *same* distance scale and we get equal-width (vs. quantile) bins that preserve the horizontal axis in units of distance, making trends and tails directly interpretable. Also the per-bin medians/IQRs are more robust to outliers. We use $m \in \{6, 8\}$ by default and bins with very small counts are reported but interpreted cautiously.

**Fairness and explanation stability.** Classical fairness metrics (e.g., demographic parity, equalized odds, counterfactual fairness) evaluate disparities in outcomes across groups [23, 47], typically distinguished into a "protected" and "unprotected" group. They do not consider whether the reasoning process leading to those outcomes is consistent. Procedural fairness foregrounds the process itself: similar individuals should be treated through a decision

mechanism that is consistent, transparent, and impartial [17, 24, 48, 49].

We operationalise a procedural stability check. For an individual $x$, small and plausible counterfactuals $z \in \mathcal{S}_K(x)$ should not radically change the explanation $\phi(x)$. Our instance-level quantity $I_2(x)$ summarises how much the attributions move under identity-preserving variation. Lower values indicate a more consistent reasoning process. At the population level, we compare the distribution of $I_2(x_i)$ across $\mathcal{X}$ within each model across groups derived from the sensitive attribute. Systematically larger instability for a minority group, particularly when the most volatile features are strong predictors of the sensitive attribute (proxies), is a potential *procedural-fairness risk signal.*

This assessment is diagnostic rather than dispositive. It does not assert normative fairness by itself. The independent masker can mischaracterise dependence structure, and instability can increase near the decision boundary even for well-behaved models. To mitigate confounding, we distance-match CFs, report boundary proximity via $|\ell(x)|$, inspect proxy features, and include coverage and $K_{\text{eff}}$ so that group comparisons are not driven by generator anomalies. Within these caveats, persistent group gaps in $I_2(x)$ together with proxy volatility provide actionable evidence of process inconsistency aligned with procedural-fairness concerns.

## 4  Experimental Setup

We study the stability of post-hoc attributions by measuring how SHAP values change between an instance $x$ and nearby, feasible counterfactuals. The goal is to assess whether the explanations remain consistent under realistic edits while controlling for in the counterfactual search.

**Datasets & preprocessing.**  We evaluate on four tabular fairness benchmarks: `Adult` (Census Income) [50], `Adult Reconstruction` [25], `Compas` (recidivism) [26], and `Dutch` [51], following prior work [27]. All datasets are split into 70% train and 30% test with stratification on the label. All preprocessing transformations are fit on the training split only and applied to the test set and the generated counterfactuals. Heavy-tailed continuous features such as `capital-gain` and `capital-loss` undergo a `log1p` transform followed by MinMax scaling. Other continuous features such as `age`, `hours-per-week`, and `educational-num` are MinMax scaled while categorical variables are one-hot encoded. Across datasets we keep the sensitive attribute binary in the transformed feature space and apply one-hot encoding only to non-sensitive categoricals. Detailed dataset notes appear in Appendix A.3.

**Table 1.** Test classification accuracy/$F_1$/AUC [%].

| Model | Adult | Adult Rec | Compas | Dutch |
|---|---|---|---|---|
| Majority Vote | 75/00/50 | 75/00/50 | 53/00/50 | 52/00/50 |
| LR | 84/64/89 | 84/64/90 | 73/70/80 | 84/83/91 |
| MLP | 80/65/88 | 81/66/89 | 69/66/76 | 83/83/91 |
| Tree | 81/62/76 | 82/62/77 | 67/66/67 | 81/80/85 |
| Random Forest | 84/65/89 | 84/66/89 | 74/70/80 | 83/81/90 |
| HGBT | 87/70/92 | 87/71/92 | 72/69/80 | 84/83/92 |

**Models.**  We report results for a logistic regression, a small multilayer perceptron, a decision tree, a random forest (RF), and a histogram-based gradient boosting (HGBT) tree. In practice, Monte Carlo estimation of SHAP and a finite background $\mathcal{B}$ introduce small sampling error; the bound then holds up to an additional estimation term that vanishes as the number of background and evaluation samples increases. Stability statistics are interpreted within each model; any cross-model remarks concern replication of patterns rather than magnitudes. Details on the models are provided in Appendix A.1 and Table 1 reports on performance metrics.

**Attribution configuration (masker, output scale).**  We explain the model's margin or logit $\ell(x)$ rather than the probability. This avoids probability-scale compression near 0 and 1 and preserves SHAP additivity on a consistent unit. Concretely, $\ell(x) = w^\top x + b$ for logistic regression, $\ell(x)$ is the pre-sigmoid logit for multilayer perceptrons, and $\ell(x) = \text{logit}(\hat{p}(x)) = \log \frac{\hat{p}(x)}{1-\hat{p}(x)}$ for models that expose only $\hat{p}(x)$ such as standard trees.

We use interventional SHAP with an independent masker. For any coalition $S \subseteq \{1, \ldots, d\}$ the features in the complement $\bar{S}$ are imputed by drawing rows from a fixed background set $\mathcal{B}$ taken from the training split. To make explanations comparable within a dataset, we fix a single $B = |\mathcal{B}|$ and reuse it across all models for that dataset. The background acts as a Monte Carlo sample for the interventional expectation. Its sampling error decreases on the order of $O(1/\sqrt{B})$ while runtime and memory grow with $B$. We form $B$ by label- and group-stratified random subsampling of the training inputs with a fixed seed and a fixed budget $B = 1000$. This ensures that $B$ is in-sample, represents both outcome classes and sensitive-attribute groups, and removes background-induced variation by reusing the same $B$. All stability analyses are conducted within each model family and we do not assume cross-model comparability of raw SHAP magnitudes.

**Counterfactual generation details (K, R).**  We randomly sample $n_{\text{candidates}} = 1000$ query instances from the test set, except for RF on `Compas`, where we only sample ten. For each query instance $x$, we generate $K_{\text{gen}} = 5$ counterfactual candidates with DiCE under $R = 5$ random seeds (yielding up to

$5 \cdot 5 = 25$ candidates per $x$), pool across seeds, deduplicate in the standardised input space, and retain the $K = 5$ nearest by standardised Euclidean distance to $x$. Unless stated otherwise, the sensitive attribute is not frozen during generation. We compute coverage (fraction with $K_{\text{eff}}(x) > 0$) and the effective set size $K_{\text{eff}}(x) = |\mathcal{S}_K(x)|$ as diagnostics to ensure results are not driven by CF search anomalies.

**Evaluation metrics and implementation details.** Instance-level instability is summarised from per-feature SHAP shifts between $x$ and its selected counterfactuals $\mathcal{S}_K(x)$ by taking the median per feature and aggregating with the $L_2$ norm. We also compute coverage as the fraction of individuals with $K_{\text{eff}}(x) \geq 1$ and the distribution of $K_{\text{eff}}(x)$. To separate generator anomalies from model behaviour we stratify by counterfactual distance, report boundary proximity through $|\ell(x)|$, measure diversity within $\mathcal{S}_K(x)$ using pairwise standardised distances, and compute a local outlier factor rate [13] on counterfactual deltas. Sampling, model training, counterfactual generation, and explanation runs all use fixed seeds per dataset and hyperparameters are held constant across datasets unless otherwise noted.

# 5 Experiments

We study whether SHAP attribution remain stable under plausible counterfactual and whether instability differs across sensitive groups once we control the proximity to the decision boundary. All results are reported within a model family since raw SHAP magnitudes are not comparable across groups. We use the multi-seed, distance controlled counterfactual set $\mathcal{S}_K(x)$ defined in Section 3. For each query point $x$, we collect multi-seed candidates, deduplicate them, and retain the $K = 5$ nearest counterfactuals. Instability for an instance is summarised by $\widetilde{\Delta}_j(x)$ and $I_2(x)$ as explained in Section 3. Lower $I_2(x)$ means a more stable explanation under plausible edits.

**Counterfactual SHAP: Sanity and Diagnosis.** We compute coverage $\Pr\left(K_{\text{eff}}(x) > 0\right)$ and the distribution of $K_{\text{eff}}(x) = |\mathcal{S}_K(x)|$ to ensure that results are not driven by search artifacts. Across datasets and models, we observe that coverage is high and the median $K_{\text{eff}}$ matches the target $K$ in both groups: median $K_{\text{eff}} = 5$ and Coverage $= 1.0$ across all dataset-model pairs. This indicates that group comparisons are not confounded by systematic failures of the CF search.

Figure 1 shows the distribution of $I_2(x)$ for each dataset-model pair using $\mathcal{S}_K(x)$ in a violin plot. These plots quantify how much explanations move under nearly feasible edits across the full evaluation set. We observe clear model-family effects. Trees

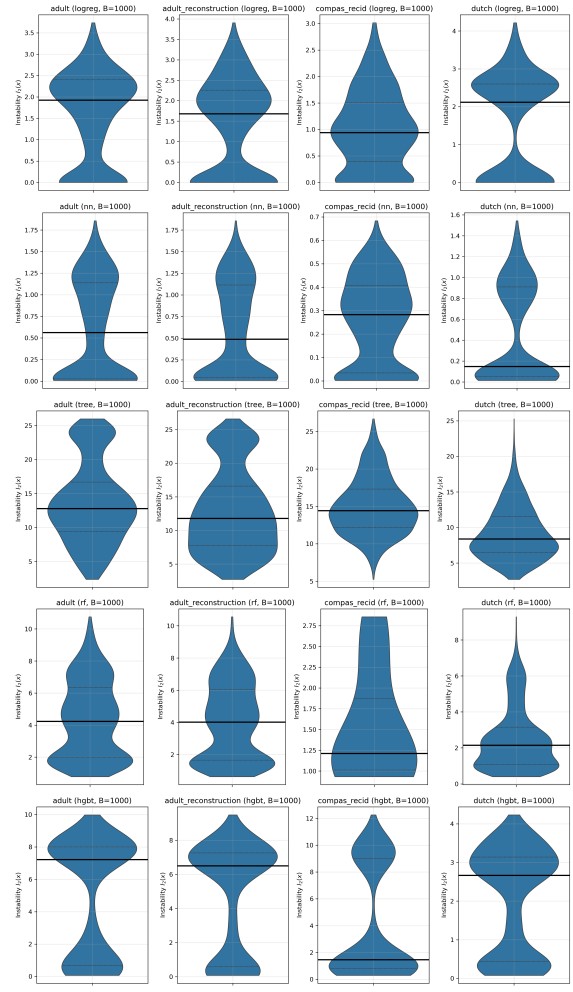

**Figure 1.** Instability $I_2$ distributions per dataset (column) and model (row).

show a higher median $I_2$ and a broader IQR than the logistic regression or the MLP, with a heavier upper tail indicating more frequent extreme instabilities. This matches the piecewise-constant structure of tree paths, where small edits can switch leaves and flip several attribution terms at once. Logistic regression exhibits intermediate stability, while the MLP is the most stable, consistent with a smoother logit in the transformed space. Dataset effects are also visible. One hot heavy datasets such as `Adult` and `Dutch` produce broader $I_2$ because many small attribution shifts accumulate in the $\ell_2$ norm. We therefore perform a proximity-conditioned analysis to separate boundary effects from other sources of instability. Search anomalies are unlikely to drive these shapes: counterfactuals are pooled across seeds, deduplicate candidates, and the $K = 5$ nearest moves are retained. Coverage and the effective set-size are high in our runs, so the distributions primarily reflect model and explainer behaviour rather than counterfactual search noise. These experiments answer the baseline question "are explanations more fragile for one group" before any boundary control.

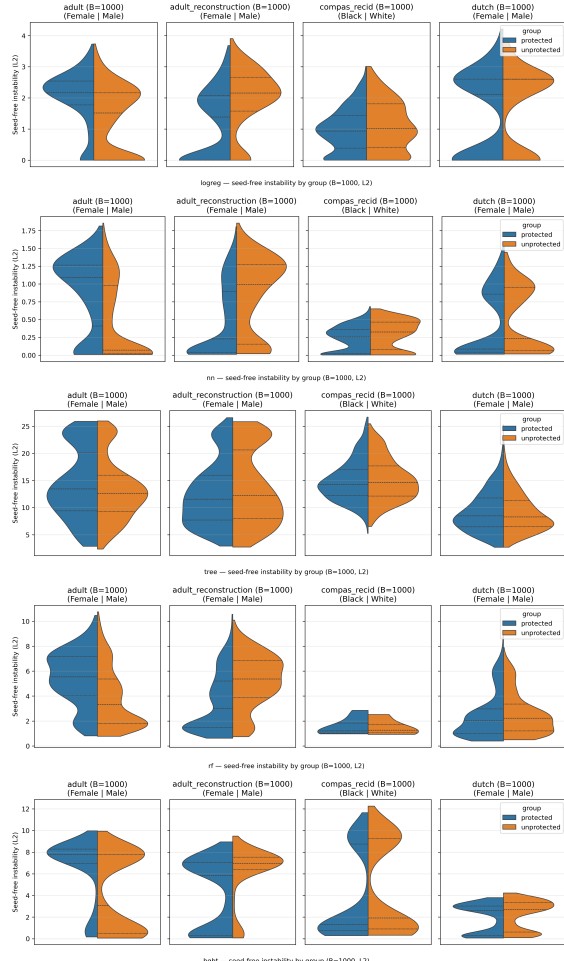

**Figure 2.** Instability $I_2$ distributions per dataset and model separated by the sensitive attribute.

We treat these as descriptive views, since instability also varies with distance to the decision boundary.

For a *distance-stability diagnosis*, we first ask whether explanation instability increases with how far the counterfactuals move an instance. Figure B.2 depicts the median $I_2(x)$ against equal-width bins of proximity (either $|\ell(x)|$ when available, or the median $L_2$ distance to the selected counterfactuals), with IQR bands for the models. Across datasets we observe a monotone increase of $I_2(x)$ with respect to the distance to the decision boundary for MLPs in Figure B.2. Analogous trends hold for the other models in the Figures B.1, B.3, B.4, and B.5.

**Groupwise instability diagnosis** Figure 2 shows, for each dataset-model pair, separated violins of the seed-pooled instability $I_2(x)$ for the protected and unprotected groups. Across models, the overall *level* and *spread* of $I_2(x)$ differs markedly: tree models exhibit the largest medians and widest violins (heavy upper tails), logistic regression is intermediate, and MLPs are the most concentrated around low values. Within datasets, visible group

gaps are present in several panels: on `Adult` and `Adult-Recon`, the protected group's violin is shifted to the right with a higher median; on `Compas` the two halves broadly overlap with only minor shifts; on `Dutch` the protected group tends to show a modest right shift for LR/MLP, while tree spreads dominate any group difference. These plots establish *empirical disparities in explanation instability at the population level*: for some dataset-model combinations one group experiences larger typical changes in SHAP under plausible edits. We next condition on proximity to the decision boundary to separate boundary-mix effects from systematic group differences.

For a *distance-stability diagnosis* across groups, we examine the curves within the same bin to detect difference in the explanation instability (Section B.2). For example, for the dataset `Compas` and MLP model, instability increases with distance, consistent with the diagnosis that larger edits change reasons more. After proximity control, the curve for the White group lies above the curve for the Black group in several mid-to-high bins, and its ribbon is also slightly higher. This indicates that, for similarly distant moves, White individuals tend to receive less stable explanations by a small but consistent margin.

**Quantifying gaps.** For a detailed analysis of group gaps, we follow the setup in Sections 3 and 4. Let $r(x) = |\ell(x)|$ if available (or the median $L_2$ distance from $x$ to the selected CF). We partition the support of $r(\cdot)$ into $M = 8$ equal-width *candidate* bins. After discarding low-support bins, let $M_{\text{supp}} \leq M$ be the number of *supported* bins, and write $x \in b$ when $x$ falls in bin $b \in \{1, \ldots, M_{\text{supp}}\}$. For each bin $b$ (see bins under Section 3), we define the median gap $\Delta_b = \text{median}\,(I_2(x)|g = 1, x \in b) - \text{median}\,(I_2(x)|g = 0, x \in b)$ where $g \in \{0, 1\}$ denotes the sensitive groups. A nonparametric 95% bootstrap confidence interval for $\Delta_b$ is obtained by resampling within bin $b$ stratified by group. We mark bin $b$ as flagged, when $\Delta_b > 0$ and the interval excludes 0, i.e., group 1 exhibits higher instability in that proximity range. We report $m/M$, the number of flagged bins out of $M$, omitting bins with very small per-group counts treated as low-support. To aggregate across proximity while removing level differences, we subtract a bin fixed effect from each observation $\tilde{I}_2(x_i) = I_2(x_i) - \text{median}\big(I_2(x) \mid x \in b(x_i)\big)$, where $b(x_i)$ is the containing $(x_i)$. The global residual gap is the median difference between groups $\hat{\Delta} = \text{median}\big(\tilde{I}_2(x_i) \mid g_i = 1\big) - \text{median}\big(\tilde{I}_2(x_i) \mid g_i = 0\big)$, with 95% stratified bootstrap confidence interval (resampling within bins). A confidence interval excluding 0 indicates a residual group effect beyond proximity. On the proximity-controlled residuals $\tilde{I}_2(x_i)$ we compute Cliff's $\delta$.

We illustrate how to read the results in Table 2 with two representative pairs. For the `Adult` dataset

**Table 2.** Proximity-matched group gaps counts bins where $\Delta_b > 0$ and the bootstrap CI excludes 0 (group 1 above group 0). The global gap uses residuals after removing per-bin medians.

| Dataset-Model | flagged_bins | global_gap | ci_lo | ci_hi | cliffs_delta |
|---|---|---|---|---|---|
| **Adult–LR** | **0/6** | **0.000** | **-0.020** | **0.000** | **-0.034** |
| Adult–NN | 0/5 | -0.172 | -0.245 | -0.121 | -0.305 |
| Adult–Tree | 0/6 | -0.921 | -1.739 | -0.300 | -0.117 |
| Adult–RF | 0/5 | -0.476 | -0.703 | -0.188 | -0.159 |
| Adult–HGBT | 0/5 | -0.469 | -0.640 | -0.296 | -0.234 |
| Adult Rec–LR | 0/7 | -0.219 | -0.336 | -0.169 | -0.040 |
| Adult Rec–NN | 0/7 | -0.118 | -0.158 | -0.077 | -0.271 |
| Adult Rec–Tree | 0/7 | -0.635 | -1.948 | 0.089 | -0.108 |
| Adult Rec–RF | 0/5 | -0.804 | -0.991 | -0.497 | -0.251 |
| Adult Rec–HGBT | 0/5 | -0.550 | -0.737 | -0.426 | -0.298 |
| Compas–LR | 0/6 | 0.000 | -0.027 | 0.024 | 0.024 |
| Compas–NN | 3/7 | 0.003 | -0.001 | 0.009 | 0.038 |
| Compas–Tree | 0/5 | 0.262 | -0.256 | 0.738 | 0.033 |
| Compas–RF | 0/0 | -0.321 | -0.547 | 0.226 | -0.583 |
| **Compas-HGBT** | **1/6** | **0.279** | **0.106** | **0.507** | **0.127** |
| Dutch–LR | 0/3 | 0.000 | 0.000 | 0.000 | -0.109 |
| Dutch–NN | 0/3 | -0.041 | -0.052 | -0.030 | -0.285 |
| Dutch–Tree | 0/7 | 0.106 | -0.343 | 0.677 | 0.000 |
| Dutch–RF | 0/6 | -0.069 | -0.214 | 0.033 | -0.030 |
| Dutch–HGBT | 0/3 | -0.181 | -0.341 | -0.103 | -0.196 |

with logistic regression, no proximity bin is flagged (0/6). The global residual gap is 0.000 with 95% confidence interval $[-0.020, 0.000]$, which includes 0, and Cliff's $\delta = -0.034$ showing that the overall difference favours group 1 slightly (negative means group 1 is more stable) and the interval touches 0. Therefore, we find no evidence of group-specific instability. The near-zero negative sign suggests that group 1 may be more stable, but this effect is not statistically distinguishable from zero.

In contrast, for the `Compas` dataset with the histogram-based gradient boosting tree (HGBT), one out of six supported proximity bins shows that group 1 (White) has significantly higher median instability than group 0 (Black). The global residual gap is 0.279 with a 95% CI of $[0.106, 0.507]$, indicating a positive gap that is statistically different from 0. Therefore, group 1 has less stable explanations overall. Cliff's $\delta = 0.127$ indicates a small effect size. Thus, for this dataset-model pair, there is a small but consistent procedural-fairness risk signal.

**Takeaways.** (i) Instability increases with distance to the decision boundary across models and datasets, with trees consistently exhibiting higher instability.

(ii) We identify actionable procedural-fairness risk signals in specific dataset-model pairs (e.g., `Compas-MLP`), where, after proximity matching, one group's explanations are systematically less stable. Other pairs show no residual gap, indicating no flag.

(iii) Coverage and effective set size are high and comparable across groups, making search anomalies an unlikely explanation for the observed patterns.

## 6 Limitations

**Diagnostic, not normative.** Our audit produces a *procedural-fairness risk signal* from explanation stability; it does not by itself establish normative (legal or ethical) unfairness. High $I_2$ or protected-reference gaps indicate process inconsistency under our setup, but interpretation should consider context, domain policy, and stakes.

**Dependence on counterfactual generation.** We probe stability using DiCE-generated edits that flip the decision under user-specified feasibility constraints. Although we pool across seeds and deduplicate, results still depend on the constraint set (mutable features, bounds, sparsity), DiCE's search objective, hyperparameters, and the model class. If the constraint set is too loose or too tight, the induced neighbourhood may be unrealistic or empty. Coverage and $K_{\text{eff}}$ mitigate, but do not eliminate, this dependence.

**Boundary effects and conditioning.** Instability naturally increases near the decision boundary. Our equal-width binning by $|\ell(x)|$ (or CF distance proxy) reduces, but may not fully remove, boundary confounding, especially when logits saturate or are noisy. Residual gaps should be interpreted alongside per-bin counts and IQRs.

**Scope of methods.** Our analysis focuses on local, additive attributions (SHAP) and tabular models (LR, trees, MLPs). Other explanation forms (rules, concepts, counterfactual sets) or modalities may exhibit different behaviours.

## 7 Conclusion

We audited *explanation stability* by pairing interventional SHAP on the margin/logit with a *multi-seed, distance-controlled* counterfactual neighbourhood. For each instance, we pooled and deduplicated DiCE candidates, retained the $K$ nearest, and summarized instability via $I_2$ from per-feature SHAP shifts, alongside coverage, $K_{\text{eff}}$, and proximity diagnostics. Across four tabular fairness benchmarks and three model families, we observed instability increases with distance to the decision boundary. Tree-based models were generally less stable than LR/MLP; after proximity control, most dataset-model pairs showed overlapping group distributions, while a few exhibited small but systematic protected-reference gaps, i.e., *procedural-fairness risk signals* under our setup. The audit was model-agnostic and practical (batched inference, shared background) and complemented accuracy and outcome fairness by evaluating whether similar individuals receive stable explanations. In our setup, a procedural fairness risk is flagged whenever one group exhibits significantly higher explanation instability after proximity matching. This implies that, despite comparable outcomes, similar cases in that group receive less consistent reasons. As a diagnostic (rather than normative) tool, it helped prioritise remediation.

# Acknowledgements

This research is funded by the European Union (COMFORT-Computational Models FOR patienT stratification in urologic cancers – Creating robust and trustworthy multimodal AI for health care), 101079894.

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

# A  Experimental Details

## A.1  Implementation Details

We use `scikit-learn` [52] to implement almost all models: logistic regression, decision tree, random forest, and histogram-based gradient boosting (HGBT) tree. For the logistic regression model, we use the solver `liblinear` for a maximum of 100 iterations. The neural network/MLP is implemented using `PyTorch` [53]. The MLP consists of one 32-dimensional hidden layer with a ReLU activation and an output layer with a sigmoid activation. The loss function is binary cross entropy. We train the MLP for 100 epochs with a batch size of 100 using Adam with a learning rate of 1e-4 and a learning rate schedule with a $\gamma = 0.99$. To compute counterfactuals, we use the Python package `dice_ml`[2] [12] and to compute SHAP values we use the package `shap`[3] [1]. All experiments run on an Intel Xeon machine with 28 cores with 2.60 GHz and 256 GB of memory.

## A.2  Procedure

A simplified outline of our procedure is as follows.

```
1. Preprocess train set and apply the
   same transformation on test set
2. Train model on train set
3. Evaluate accuracy, F1, and AUC on
   test set
4. Build SHAP background and explainer
   using an IndependentMasker() and
   PermutationExplainer()
5. Build DICE object
6. Generate CFs across seeds and pool
7. Select K nearest CFs per instance in
    standardised input space
8. SHAP of originals and CFs
9. Aggregate to instance level (feature
   -wise median across CFs)
10. Return all metrics
```

## A.3  Dataset Details

`Adult Reconstruction` [25] simulates real-world issues where sensitive attributes and labels may suffer from measurement bias and covariate shift. The raw data contain 49,531 instances with 14 attributes. We follow the cleaned subset used in prior work, yielding 45,849 instances. The label is whether income exceeds $50,000 per year. The protected attribute is `gender`. We remove the original income column when a binary target `income_bin` is present and map text labels to $\{0, 1\}$. We keep `gender` as a binary passthrough feature and one-hot encode other categoricals.

---

[2] https://interpret.ml/DiCE/
[3] https://shap.readthedocs.io/en/

`Adult (Census Income)` [50] contains 48,842 rows with 14 demographic attributes. After removing rows with missing entries and dropping non-informative columns such as `fnlwgt`, we obtain 45,222 examples. The task is to predict income greater than $50,000. The sensitive attribute is `gender`. The male to female ratio is about 66.9% to 33.1%. We apply the same column-wise transforms as above, including `log1p` for `capital-gain` and `capital-loss`.

`Dutch (Census)` [51] contains 60,420 samples with 11 attributes describing aggregated census information. The binary task is to predict whether an individual holds a high-prestige `occupation`. The protected attribute is `sex`, which is close to balanced in this dataset. We one-hot encode categoricals and scale continuous features as described above.

`Compas (recidivism)` [26] includes 7,214 individuals with criminal history and demographic attributes. We use the cleaned `Compas` subset with 6,172 individuals and seven features, and we predict rearrest within two years `two_year_recid`. The protected attribute is `race`; the black to white ratio is approximately 51.4% to 34%. Preprocessing follows the same pattern as above. We note the known limitations of this dataset and treat it as a stress test for explanation stability rather than a gold standard of social ground truth.

**Reproducibility note.** We drop the following columns in `Compas` to prevent leakage and remove identifiers:

```
to_drop = [
  'id','name','first','last','dob',
  'c_case_number','r_case_number',
  'vr_case_number','c_charge_desc',
  'r_charge_desc','vr_charge_desc',
  'decile_score','decile_score.1',
  'v_decile_score', 'score_text',
  'v_score_text', 'type_of_assessment',
  'v_type_of_assessment','event',
  'r_days_from_arrest','r_jail_in',
  'r_jail_out', 'r_offense_date',
  'r_charge_degree', 'vr_offense_date',
  'vr_charge_degree', 'in_custody',
  'out_custody','start','end',
  'is_recid','violent_recid',
  'is_violent_recid',
  'priors_count.1','age_cat'
]
```

# B  Additional Experiments

## B.1  Runtimes

We report on the computational runtimes of our approach in Table B.1.

**Table B.1.** Computation times in seconds.

| dataset | model | B | t_train_eval | t_dice | t_build_shap | t_dice_seeds | t_fragility |
|---|---|---|---|---|---|---|---|
| adult | logreg | 1000 | 0.19 | 1328.48 | 0.0003 | 9559.66 | 52.82 |
| adult | nn | 1000 | 168.46 | 797.67 | 0.0002 | 3740.08 | 66.36 |
| adult | tree | 1000 | 0.31 | 1378.42 | 0.0003 | 6525.24 | 52.81 |
| adult | rf | 1000 | 1.94 | 20042.55 | 0.0002 | 66504.66 | 1985.51 |
| adult | hgbt | 1000 | 1.49 | 1940.14 | 0.0003 | 7122.67 | 110.38 |
| adult_reconstruction | logreg | 1000 | 0.15 | 1041.49 | 0.0002 | 4877.53 | 56.05 |
| adult_reconstruction | nn | 1000 | 183.30 | 500.08 | 0.0002 | 3046.08 | 53.97 |
| adult_reconstruction | tree | 1000 | 0.29 | 1014.38 | 0.0002 | 4530.15 | 50.18 |
| adult_reconstruction | rf | 1000 | 1.91 | 7870.75 | 0.0002 | 60351.63 | 1342.03 |
| adult_reconstruction | hgbt | 1000 | 0.88 | 1170.84 | 0.0002 | 3669.90 | 80.50 |
| compas_recid | logreg | 1000 | 0.02 | 4087.97 | 0.0003 | 20525.07 | 15.80 |
| compas_recid | nn | 1000 | 20.20 | 2230.09 | 0.0003 | 10121.34 | 18.01 |
| compas_recid | tree | 1000 | 0.04 | 4605.94 | 0.0003 | 22513.11 | 17.03 |
| compas_recid | rf | 1000 | N/A | N/A | N/A | N/A | N/A |
| compas_recid | hgbt | 1000 | 1.23 | 14306.68 | 0.0004 | 68963.22 | 110.79 |
| dutch | logreg | 1000 | 0.32 | 492.96 | 0.0002 | 2184.19 | 38.08 |
| dutch | nn | 1000 | 258.22 | 474.98 | 0.0003 | 1793.32 | 30.30 |
| dutch | tree | 1000 | 0.26 | 564.52 | 0.0002 | 2787.92 | 31.77 |
| dutch | rf | 1000 | 2.01 | 13578.26 | 0.0002 | 50287.27 | 1757.33 |
| dutch | hgbt | 1000 | 0.93 | 730.18 | 0.0002 | 4819.22 | 69.00 |

## B.2 Instability of $I_2$ per Dataset

For completeness, we now provide all instability $I_2$ results for all datasets and models. Figures B.1, B.2, B.3, B.4, and B.5 show the results pooled among all data points whereas Figures B.6, B.7, B.8, B.9 and B.10 depict the results separated by the sensitive attribute.

## B.3 Evaluating the Impact of $B$

Within the paper, we used a fixed background size of $B = 1000$. To assess sensitivity, we vary the independent-masker background $B \in \{5, 10, 50, 100, 500, 10000\}$, while keeping the model, seeds, CF selection, and preprocessing fixed. For the Figures B.11, B.12, B.13, and B.14 we recomputed the instability $I_2$, while keeping everything else fixed. Across datasets and models, the distribution of $I_2$ is essentially unchanged as $B$ grows. E.g., for logistic regression, the median and IQR is visually indistinguishable across all $B$ and datasets. However, for the multi-layer perceptron on the `Dutch` dataset, we observe noticeable sensitivity for small background-sizes $B \leq 100$.

Next, we discover how varying the independent-masker background $B \in \{5, 10, 50, 100, 500, 1000, 10000\}$ changes our proximity-matched conclusions in Table B.2, B.3, B.4, and B.5. We observe that no dataset–model pair gains or loses a significant residual gap when $B$ changes.

Overall we can observe, that $B = 1000$ is a safe default. Typically $B \in [300, 500]$ is sufficient, while $B \leq 100$ can add variance.

## B.4 Normalised Instability of $I_2$ per Dataset

Finally, we consider the normalised instability and obtain the following results Figure B.15. Across all datasets, logistic regression and the MLP remain the most stable. Tree based models are broader with heavier upper tails even after normalisation.

**Table B.2.** Proximity-matched group gaps for the Adult dataset counts bins where $\Delta_b > 0$ and the bootstrap CI excludes 0 (group 1 above group 0). The global gap uses residuals after removing per-bin medians.

| Dataset-Model | B | flagged_bins | global_gap | ci_lo | ci_hi | cliffs_delta |
|---|---|---|---|---|---|---|
| Adult–LR | 5 | 0/6 | 0.000 | -0.020 | 0.000 | -0.037 |
| Adult–LR | 10 | 0/6 | 0.000 | -0.022 | 0.000 | -0.037 |
| Adult–LR | 50 | 0/6 | 0.000 | -0.017 | 0.000 | -0.034 |
| Adult–LR | 100 | 0/6 | 0.000 | -0.020 | 0.000 | -0.035 |
| Adult–LR | 500 | 0/6 | 0.000 | -0.020 | 0.000 | -0.035 |
| Adult–LR | 10000 | 0/6 | 0.000 | -0.020 | 0.000 | -0.035 |
| Adult–NN | 5 | 0/5 | -0.219 | -0.279 | -0.172 | -0.295 |
| Adult–NN | 10 | 0/6 | -0.137 | -0.181 | -0.078 | -0.292 |
| Adult–NN | 50 | 0/5 | -0.164 | -0.220 | -0.119 | -0.322 |
| Adult–NN | 100 | 0/6 | -0.200 | -0.261 | -0.151 | -0.326 |
| Adult–NN | 500 | 0/6 | -0.198 | -0.236 | -0.135 | -0.364 |
| Adult–NN | 10000 | 0/5 | -0.235 | -0.307 | -0.185 | -0.380 |
| Adult–Tree | 5 | 0/6 | -1.062 | -1.951 | -0.247 | -0.088 |
| Adult–Tree | 10 | 0/6 | -1.019 | -1.765 | -0.252 | -0.128 |
| Adult–Tree | 50 | 0/6 | -0.903 | -1.561 | -0.266 | -0.089 |
| Adult–Tree | 100 | 0/6 | -1.179 | -2.026 | -0.424 | -0.085 |
| Adult–Tree | 500 | 0/6 | -0.627 | -1.710 | 0.036 | -0.083 |
| Adult–Tree | 10000 | 0/6 | -0.800 | -1.633 | -0.101 | -0.080 |
| Adult–RF | 5 | 0/6 | -0.830 | -1.095 | -0.430 | -0.202 |
| Adult–RF | 10 | 0/6 | -0.524 | -0.831 | -0.288 | -0.176 |
| Adult–RF | 100 | 0/6 | -0.444 | -0.652 | -0.252 | -0.157 |
| Adult–RF | 500 | 0/5 | -0.528 | -0.744 | -0.250 | -0.166 |
| Adult–HGBT | 5 | 0/5 | -0.465 | -0.648 | -0.343 | -0.240 |
| Adult–HGBT | 10 | 0/5 | -0.469 | -0.619 | -0.298 | -0.231 |
| Adult–HGBT | 50 | 0/5 | -0.510 | -0.690 | -0.317 | -0.241 |
| Adult–HGBT | 100 | 0/5 | -0.451 | -0.673 | -0.298 | -0.242 |
| Adult–HGBT | 500 | 0/5 | -0.431 | -0.648 | -0.270 | -0.234 |
| Adult–HGBT | 10000 | 0/5 | -0.428 | -0.670 | -0.263 | -0.241 |

**Table B.3.** Proximity-matched group gaps for the Adult Rec dataset counts bins where $\Delta_b > 0$ and the bootstrap CI excludes 0 (group 1 above group 0). The global gap uses residuals after removing per-bin medians.

| Dataset-Model | B | flagged_bins | global_gap | ci_lo | ci_hi | cliffs_delta |
|---|---|---|---|---|---|---|
| Adult Rec–LR | 5 | 0/7 | -0.219 | -0.342 | -0.169 | -0.043 |
| Adult Rec–LR | 10 | 0/7 | -0.219 | -0.342 | -0.169 | -0.042 |
| Adult Rec–LR | 50 | 0/7 | -0.219 | -0.342 | -0.161 | -0.041 |
| Adult Rec–LR | 100 | 0/7 | -0.219 | -0.342 | -0.169 | -0.043 |
| Adult Rec–LR | 500 | 0/7 | -0.219 | -0.336 | -0.169 | -0.040 |
| Adult Rec–LR | 10000 | 0/7 | -0.219 | -0.336 | -0.169 | -0.040 |
| Adult Rec–NN | 5 | 0/7 | -0.133 | -0.163 | -0.092 | -0.260 |
| Adult Rec–NN | 10 | 0/6 | -0.136 | -0.178 | -0.055 | -0.187 |
| Adult Rec–NN | 50 | 0/7 | -0.133 | -0.158 | -0.084 | -0.294 |
| Adult Rec–NN | 100 | 0/6 | -0.135 | -0.173 | -0.104 | -0.315 |
| Adult Rec–NN | 500 | 0/6 | -0.139 | -0.216 | -0.073 | -0.258 |
| Adult Rec–NN | 10000 | 0/6 | -0.237 | -0.318 | -0.185 | -0.308 |
| Adult Rec–Tree | 5 | 0/7 | -1.086 | -1.687 | -0.604 | -0.187 |
| Adult Rec–Tree | 10 | 0/7 | -1.185 | -2.065 | -0.495 | -0.178 |
| Adult Rec–Tree | 50 | 0/7 | -1.622 | -2.473 | -0.165 | -0.143 |
| Adult Rec–Tree | 100 | 0/7 | -1.311 | -2.154 | -0.456 | -0.141 |
| Adult Rec–Tree | 500 | 0/7 | -1.075 | -2.417 | -0.233 | -0.163 |
| Adult Rec–Tree | 10000 | 0/7 | -1.080 | -2.088 | -0.217 | -0.134 |
| Adult Rec–RF | 5 | 0/5 | -0.752 | -1.026 | -0.460 | -0.270 |
| Adult Rec–RF | 10 | 0/5 | -0.831 | -1.165 | -0.607 | -0.289 |
| Adult Rec–RF | 50 | 0/5 | -0.728 | -1.048 | -0.508 | -0.292 |
| Adult Rec–RF | 100 | 0/5 | -0.744 | -1.145 | -0.536 | -0.279 |
| Adult Rec–RF | 500 | 0/5 | -0.808 | -1.260 | -0.565 | -0.278 |
| Adult Rec–HGBT | 5 | 0/5 | -0.624 | -0.788 | -0.480 | -0.311 |
| Adult Rec–HGBT | 10 | 0/5 | -0.633 | -0.812 | -0.469 | -0.308 |
| Adult Rec–HGBT | 50 | 0/5 | -0.548 | -0.747 | -0.426 | -0.304 |
| Adult Rec–HGBT | 100 | 0/5 | -0.595 | -0.778 | -0.462 | -0.307 |
| Adult Rec–HGBT | 500 | 0/5 | -0.598 | -0.763 | -0.455 | -0.303 |
| Adult Rec–HGBT | 10000 | 0/5 | -0.591 | -0.738 | -0.454 | -0.312 |

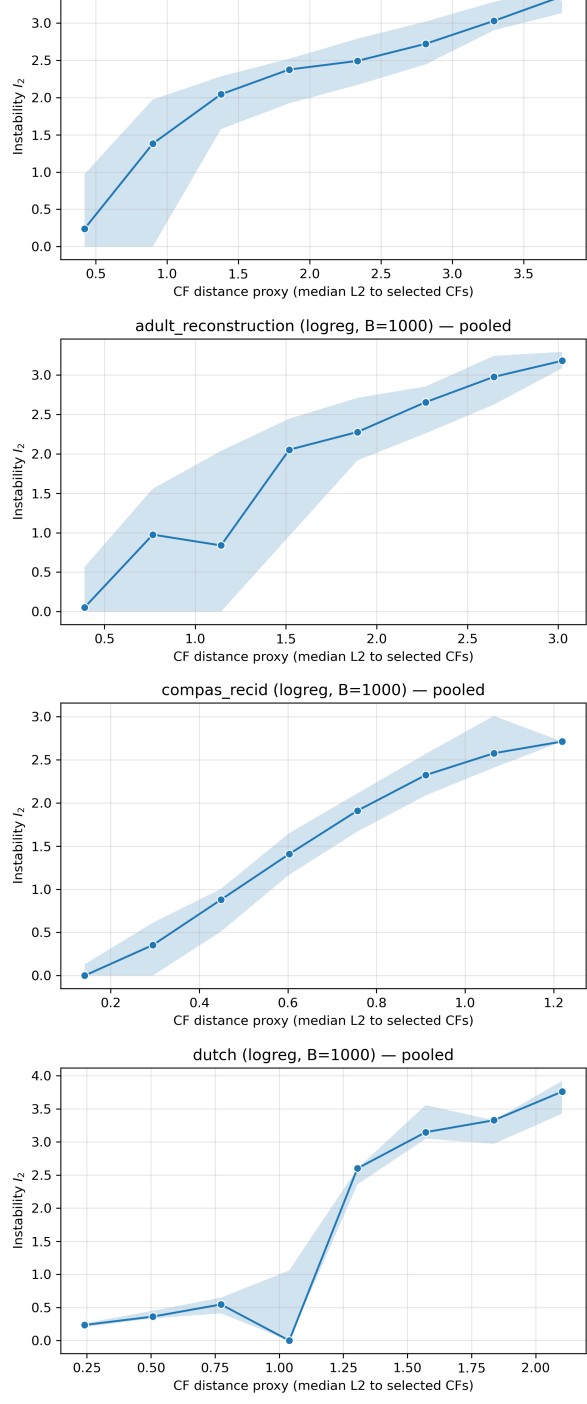

**Figure B.1.** Instability $I_2$ per dataset for the logistic regression model.

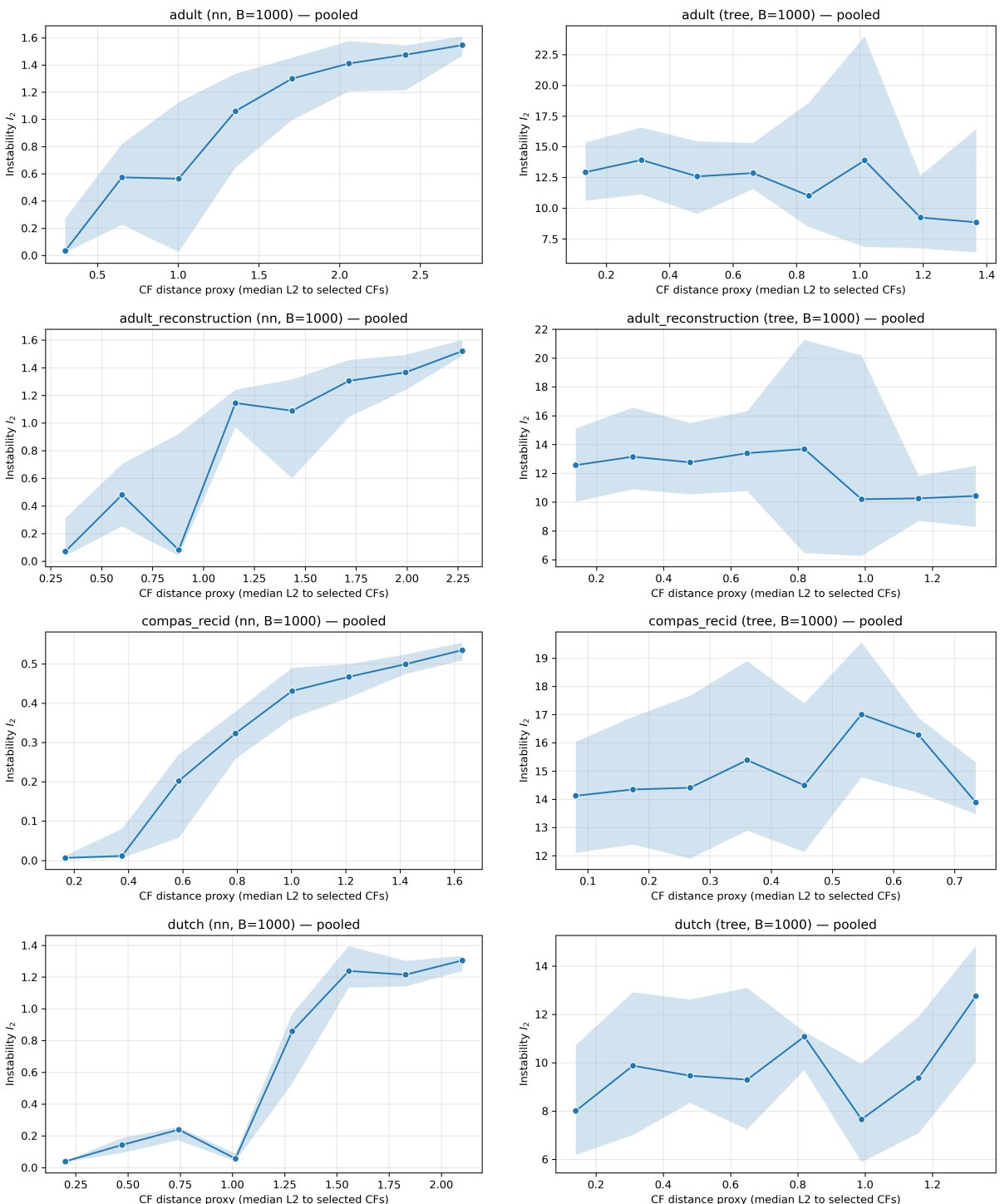

**Figure B.2.** Instability $I_2$ per dataset for the multi-layer perceptron.

**Figure B.3.** Instability $I_2$ per dataset for the tree model.

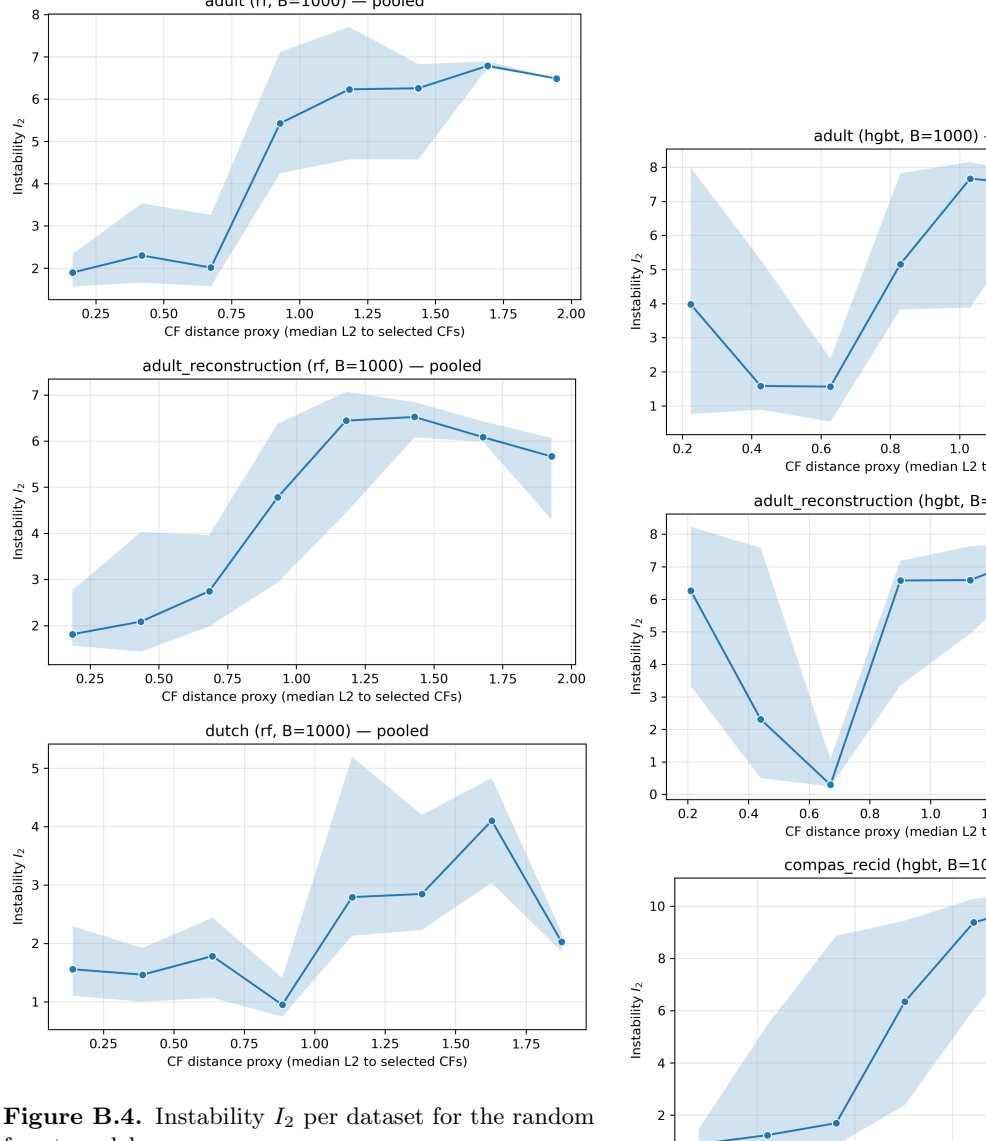

**Figure B.4.** Instability $I_2$ per dataset for the random forest model.

**Table B.4.** Proximity-matched group gaps for the Compas dataset counts bins where $\Delta_b > 0$ and the bootstrap CI excludes 0 (group 1 above group 0). The global gap uses residuals after removing per-bin medians.

| Dataset-Model | B | flagged_bins | global_gap | ci_lo | ci_hi | cliffs_delta |
|---|---|---|---|---|---|---|
| Compas–LR | 5 | 0/6 | 0.000 | -0.026 | 0.026 | 0.025 |
| Compas–LR | 10 | 0/6 | 0.000 | -0.024 | 0.024 | 0.025 |
| Compas–LR | 50 | 0/6 | 0.000 | -0.026 | 0.026 | 0.025 |
| Compas–LR | 100 | 0/6 | 0.000 | -0.026 | 0.024 | 0.025 |
| Compas–LR | 500 | 0/6 | 0.000 | -0.026 | 0.026 | 0.024 |
| Compas–NN | 5 | 3/6 | 0.019 | 0.009 | 0.049 | 0.168 |
| Compas–NN | 10 | 1/6 | 0.005 | -0.003 | 0.022 | 0.092 |
| Compas–NN | 50 | 1/7 | 0.002 | -0.003 | 0.011 | 0.022 |
| Compas–NN | 100 | 3/6 | 0.033 | 0.005 | 0.044 | 0.138 |
| Compas–NN | 500 | 1/6 | 0.001 | -0.004 | 0.010 | 0.046 |
| Compas–Tree | 5 | 0/5 | -0.425 | -1.068 | 0.110 | 0.013 |
| Compas–Tree | 10 | 0/5 | -0.188 | -0.753 | 0.414 | 0.007 |
| Compas–Tree | 50 | 0/5 | -0.009 | -0.599 | 0.514 | 0.012 |
| Compas–Tree | 100 | 0/5 | 0.176 | -0.455 | 0.661 | 0.024 |
| Compas–Tree | 500 | 0/5 | 0.152 | -0.372 | 0.669 | 0.009 |
| Compas–HGBT | 5 | 1/6 | 0.098 | -0.024 | 0.263 | 0.089 |
| Compas–HGBT | 10 | 1/6 | 0.143 | 0.000 | 0.296 | 0.089 |
| Compas–HGBT | 100 | 2/6 | 0.269 | 0.105 | 0.480 | 0.125 |
| Compas–HGBT | 500 | 1/6 | 0.281 | 0.106 | 0.486 | 0.125 |

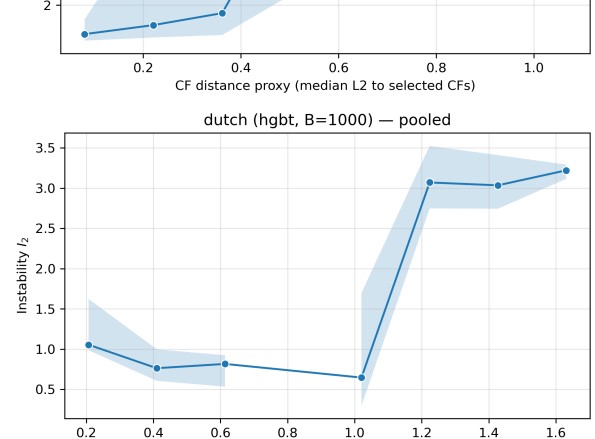

**Figure B.5.** Instability $I_2$ per dataset for the Histogram-based Gradient Boosting tree.

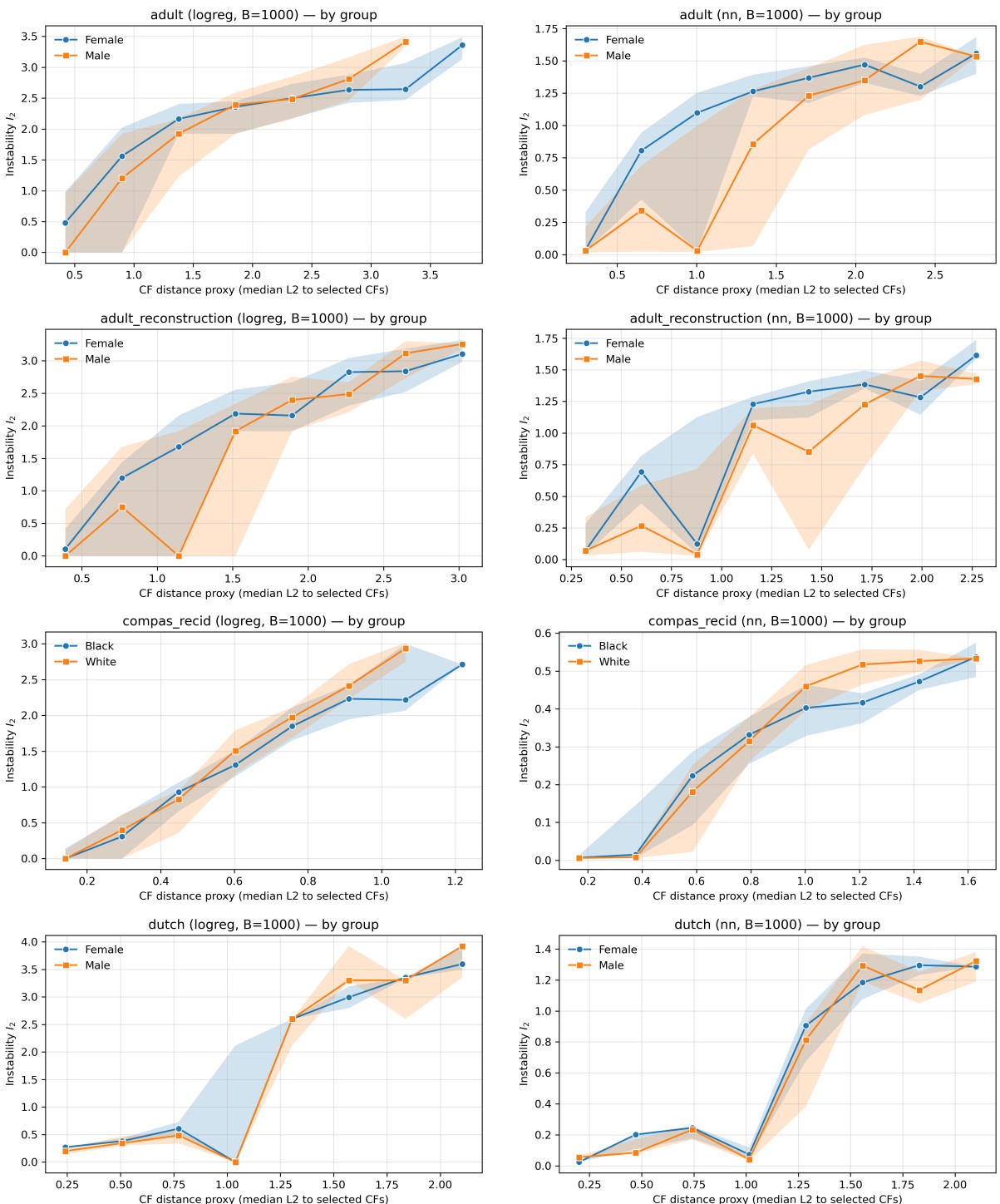

**Figure B.6.** Instability $I_2$ separated by sensitive attribute per dataset for the logistic regression model.

**Figure B.7.** Instability $I_2$ separated by sensitive attribute per dataset for the multi-layer perceptron.

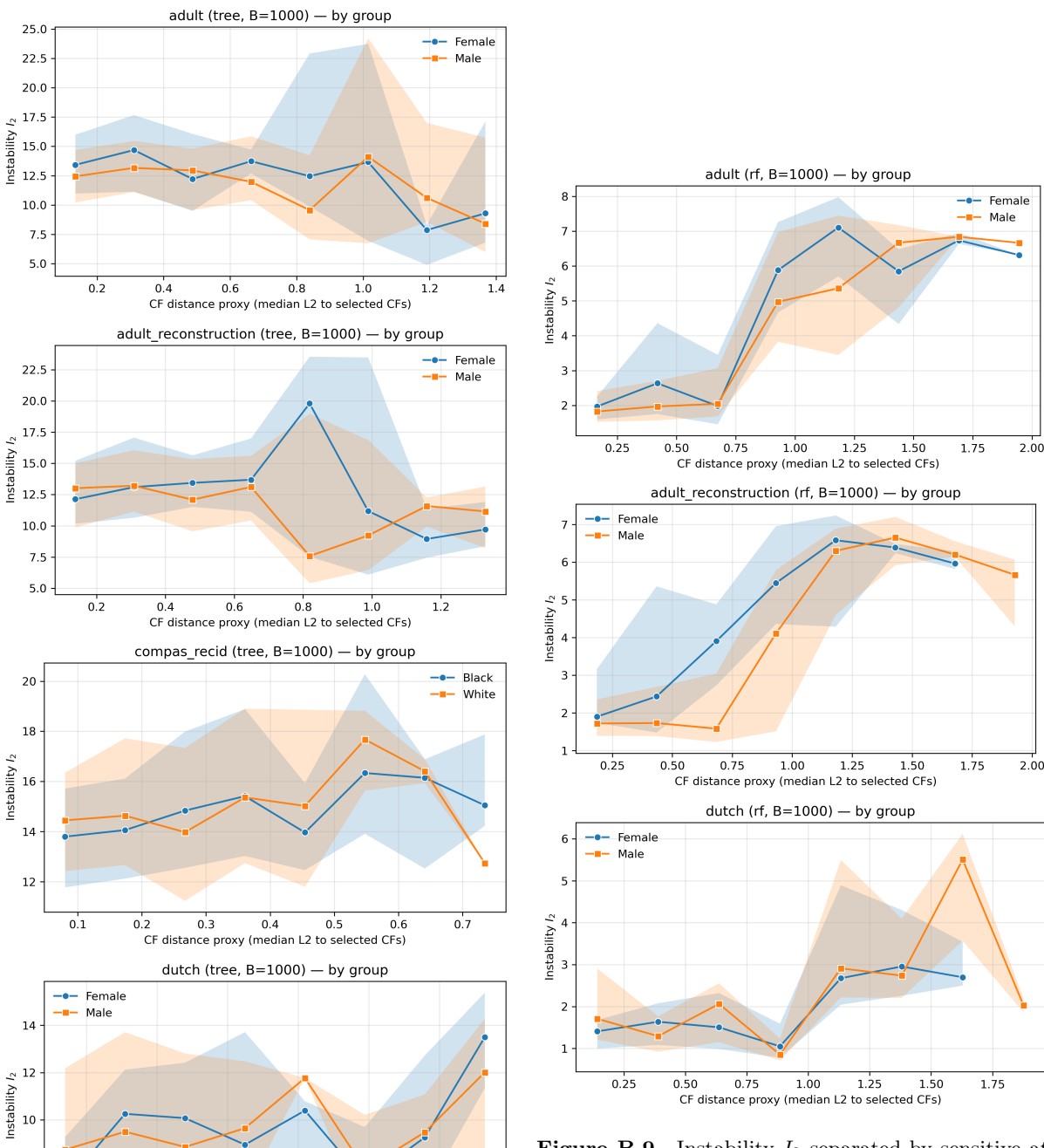

**Figure B.8.** Instability $I_2$ separated by sensitive attribute per dataset for the tree model.

**Figure B.9.** Instability $I_2$ separated by sensitive attribute per dataset for the random forest model.

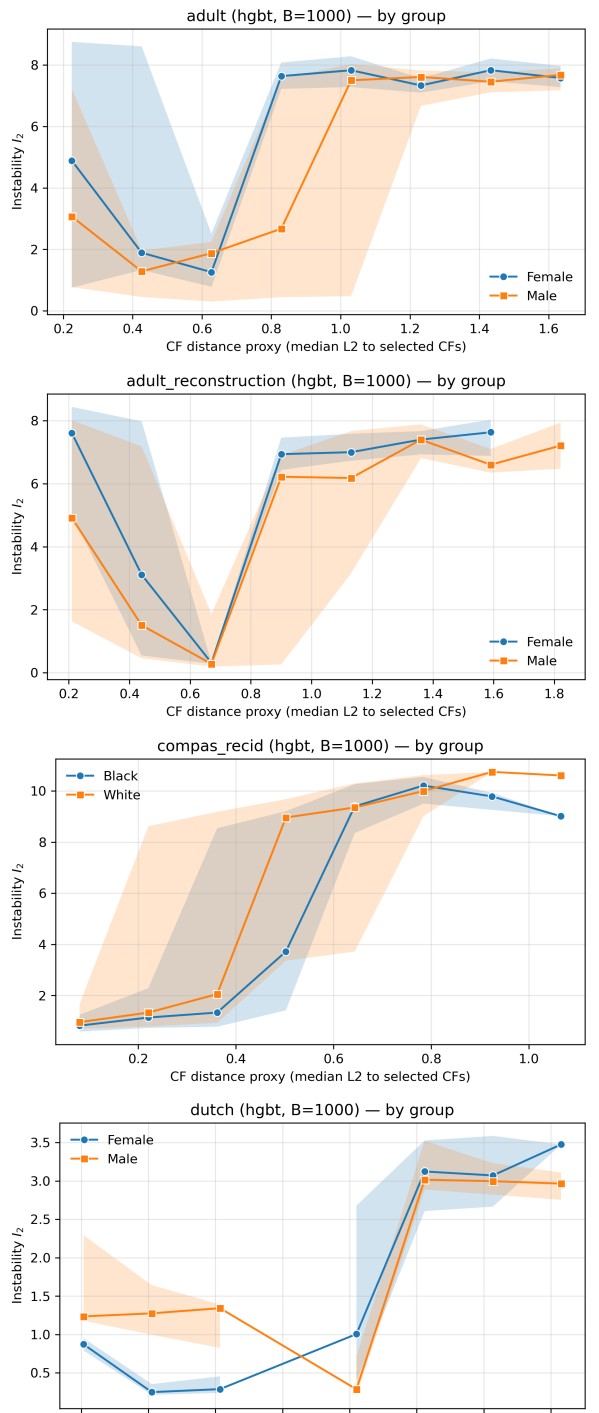

**Figure B.10.** Instability $I_2$ separated by sensitive attribute per dataset for the Histogram-based Gradient Boosting tree.

**Table B.5.** Proximity-matched group gaps for the Dutch dataset counts bins where $\Delta_b > 0$ and the bootstrap CI excludes 0 (group 1 above group 0). The global gap uses residuals after removing per-bin medians.

| Dataset-Model | B | flagged_bins | global_gap | ci_lo | ci_hi | cliffs_delta |
|---|---|---|---|---|---|---|
| dutch–LR | 5 | 0/3 | 0.000 | 0.000 | 0.000 | -0.095 |
| dutch–LR | 10 | 0/3 | 0.000 | 0.000 | 0.000 | -0.063 |
| dutch–LR | 50 | 0/3 | 0.000 | 0.000 | 0.000 | -0.081 |
| dutch–LR | 100 | 0/3 | 0.000 | 0.000 | 0.000 | -0.091 |
| dutch–LR | 500 | 0/3 | 0.000 | 0.000 | 0.000 | -0.081 |
| dutch–LR | 10000 | 0/3 | 0.000 | 0.000 | 0.000 | -0.083 |
| dutch–NN | 5 | 0/4 | -0.026 | -0.032 | -0.020 | -0.306 |
| dutch–NN | 10 | 0/4 | -0.058 | -0.073 | -0.048 | -0.425 |
| dutch–NN | 50 | 0/4 | -0.039 | -0.050 | -0.030 | -0.293 |
| dutch–NN | 100 | 0/4 | -0.053 | -0.061 | -0.046 | -0.380 |
| dutch–NN | 500 | 0/3 | -0.046 | -0.059 | -0.032 | -0.277 |
| dutch–NN | 10000 | 0/3 | -0.044 | -0.052 | -0.036 | -0.354 |
| dutch–Tree | 5 | 1/7 | 0.721 | 0.129 | 1.239 | 0.113 |
| dutch–Tree | 10 | 0/7 | -0.005 | -0.523 | 0.446 | 0.007 |
| dutch–Tree | 50 | 1/7 | 0.709 | 0.096 | 1.247 | 0.055 |
| dutch–Tree | 100 | 1/7 | 0.101 | -0.267 | 0.555 | 0.031 |
| dutch–Tree | 500 | 0/7 | -0.087 | -0.530 | 0.394 | -0.060 |
| dutch–Tree | 10000 | 1/7 | 0.391 | -0.083 | 0.936 | 0.063 |
| dutch–RF | 5 | 0/6 | -0.032 | -0.165 | 0.062 | -0.016 |
| dutch–RF | 10 | 0/6 | -0.077 | -0.218 | -0.004 | -0.047 |
| dutch–RF | 50 | 0/6 | 0.007 | -0.059 | 0.110 | 0.024 |
| dutch–RF | 100 | 0/6 | -0.020 | -0.137 | 0.089 | -0.002 |
| dutch–RF | 500 | 0/6 | -0.100 | -0.205 | -0.021 | -0.050 |
| dutch–HGBT | 5 | 0/3 | -0.195 | -0.256 | -0.122 | -0.248 |
| dutch–HGBT | 10 | 0/3 | -0.205 | -0.303 | -0.129 | -0.256 |
| dutch–HGBT | 50 | 0/3 | -0.144 | -0.274 | -0.073 | -0.170 |
| dutch–HGBT | 100 | 0/3 | -0.154 | -0.281 | -0.086 | -0.192 |
| dutch–HGBT | 500 | 0/3 | -0.222 | -0.324 | -0.146 | -0.224 |
| dutch–HGBT | 10000 | 0/3 | -0.225 | -0.295 | -0.178 | -0.242 |

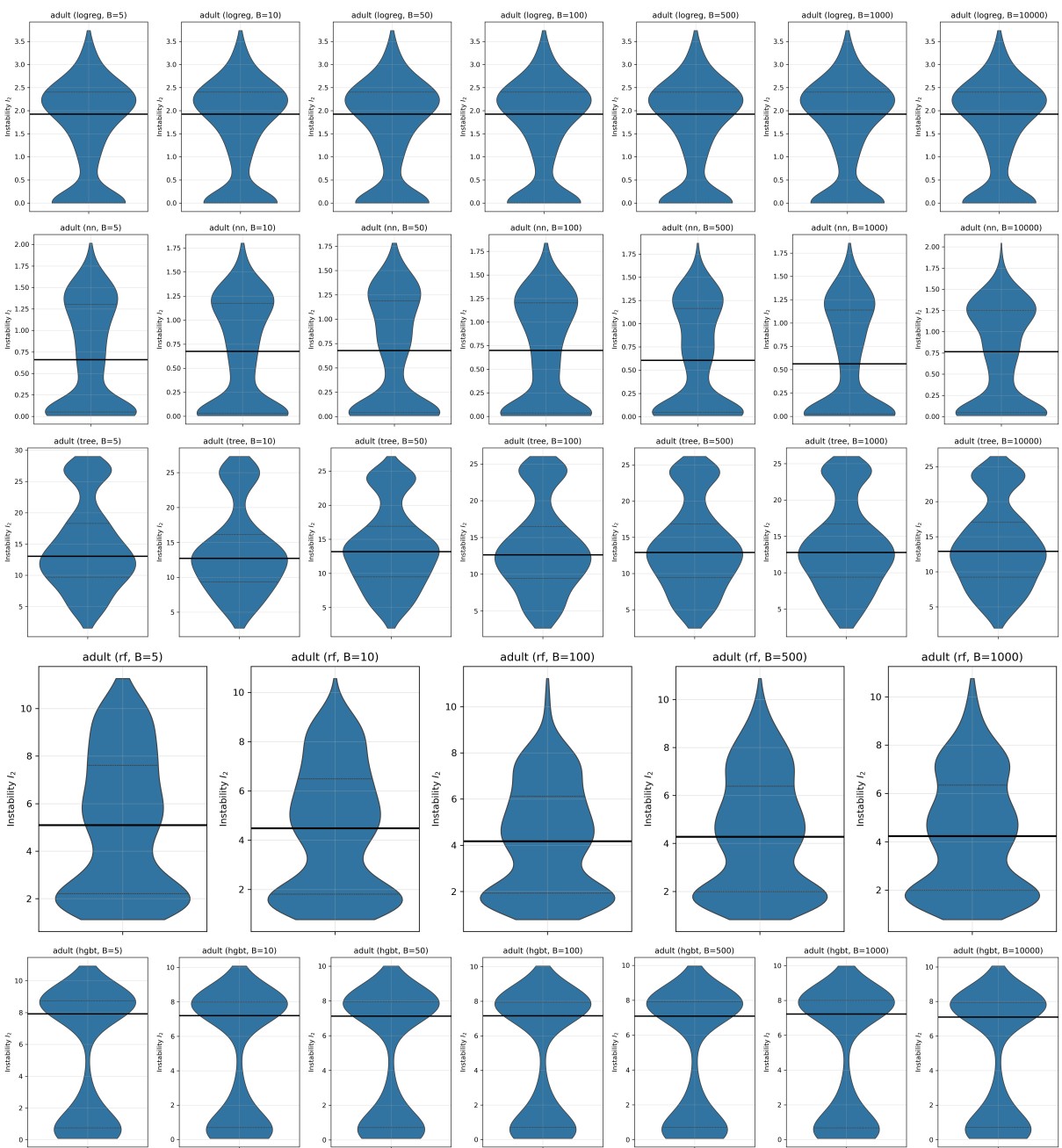

**Figure B.11.** Instability $I_2$ distributions per model and $B$ on `Adult`.

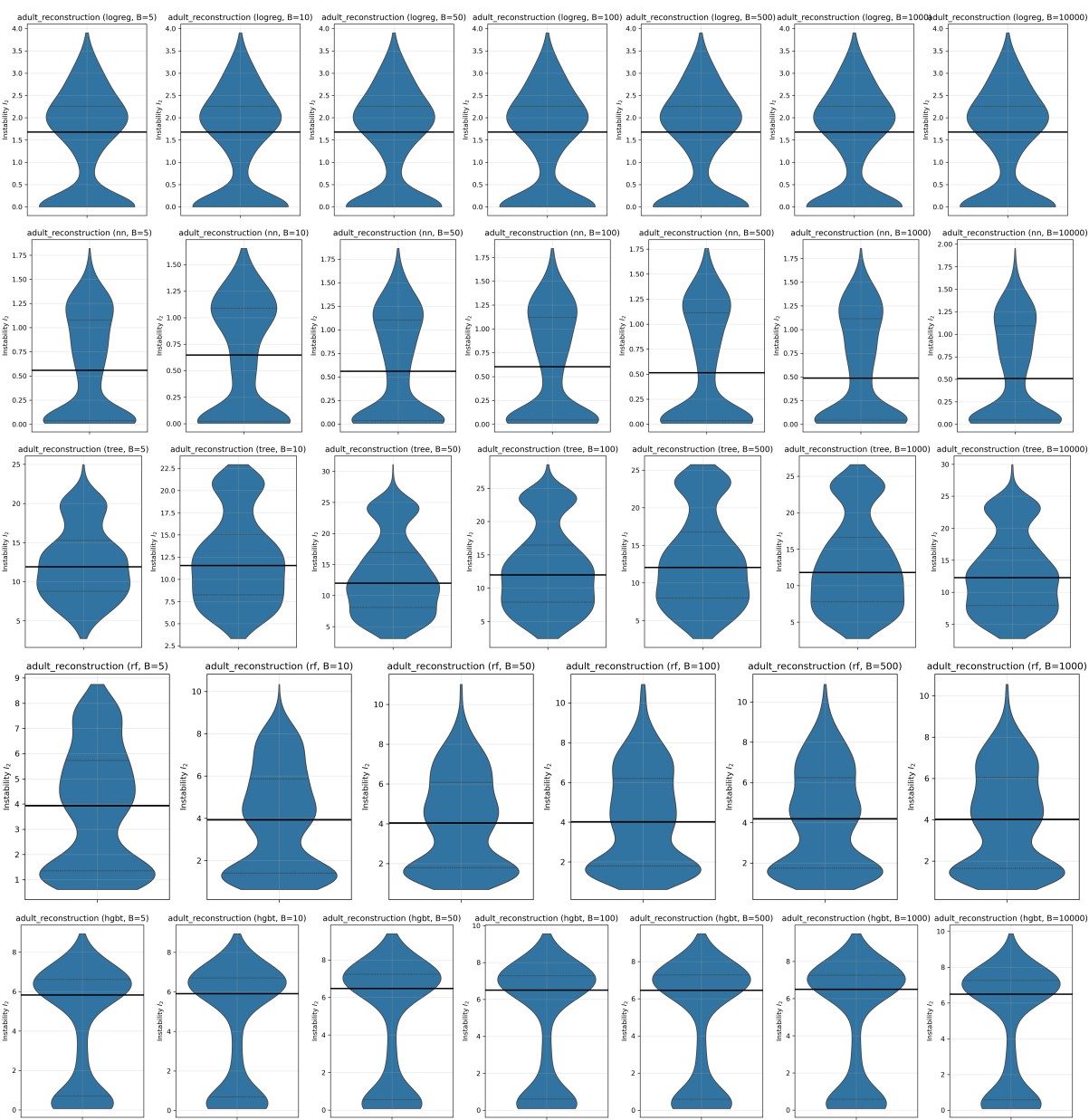

**Figure B.12.** Instability $I_2$ distributions per model and $B$ on `Adult Reconstruction`.

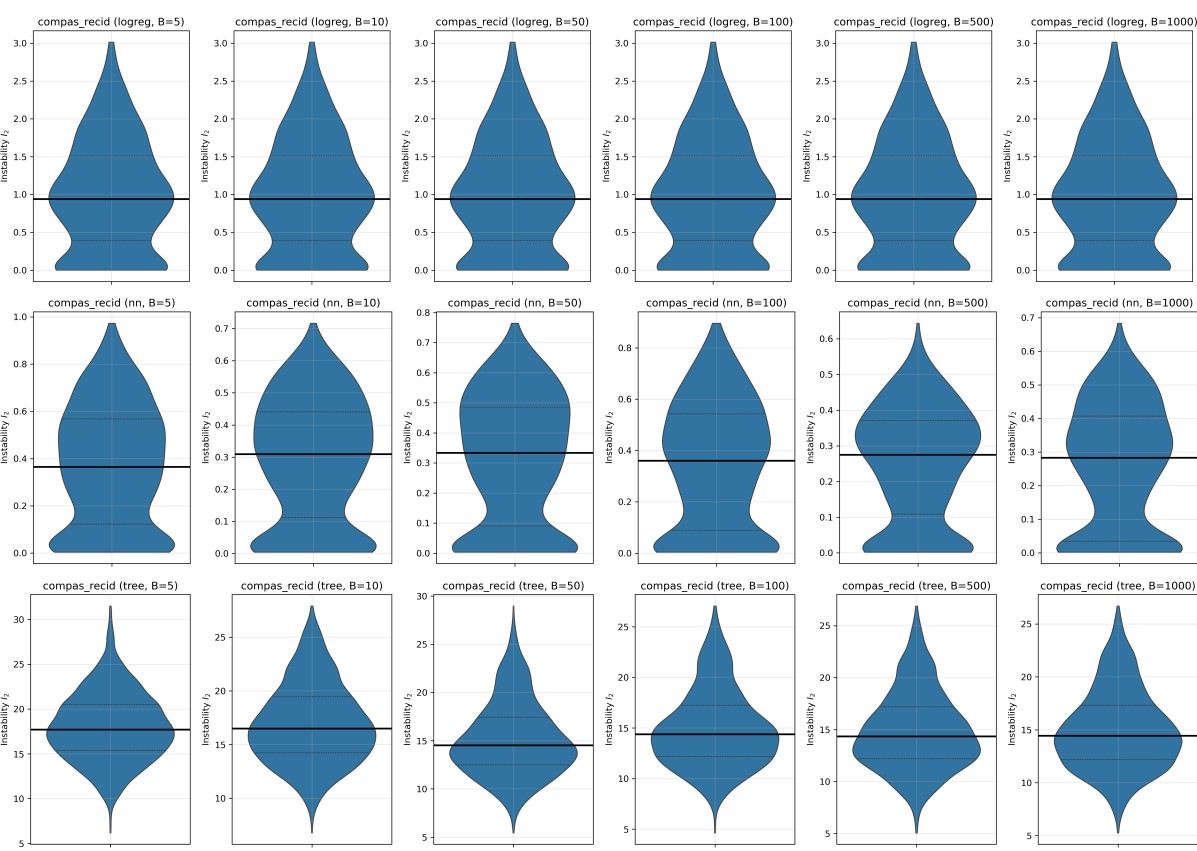

**Figure B.13.** Instability $I_2$ distributions per model and $B$ on `Compas`.

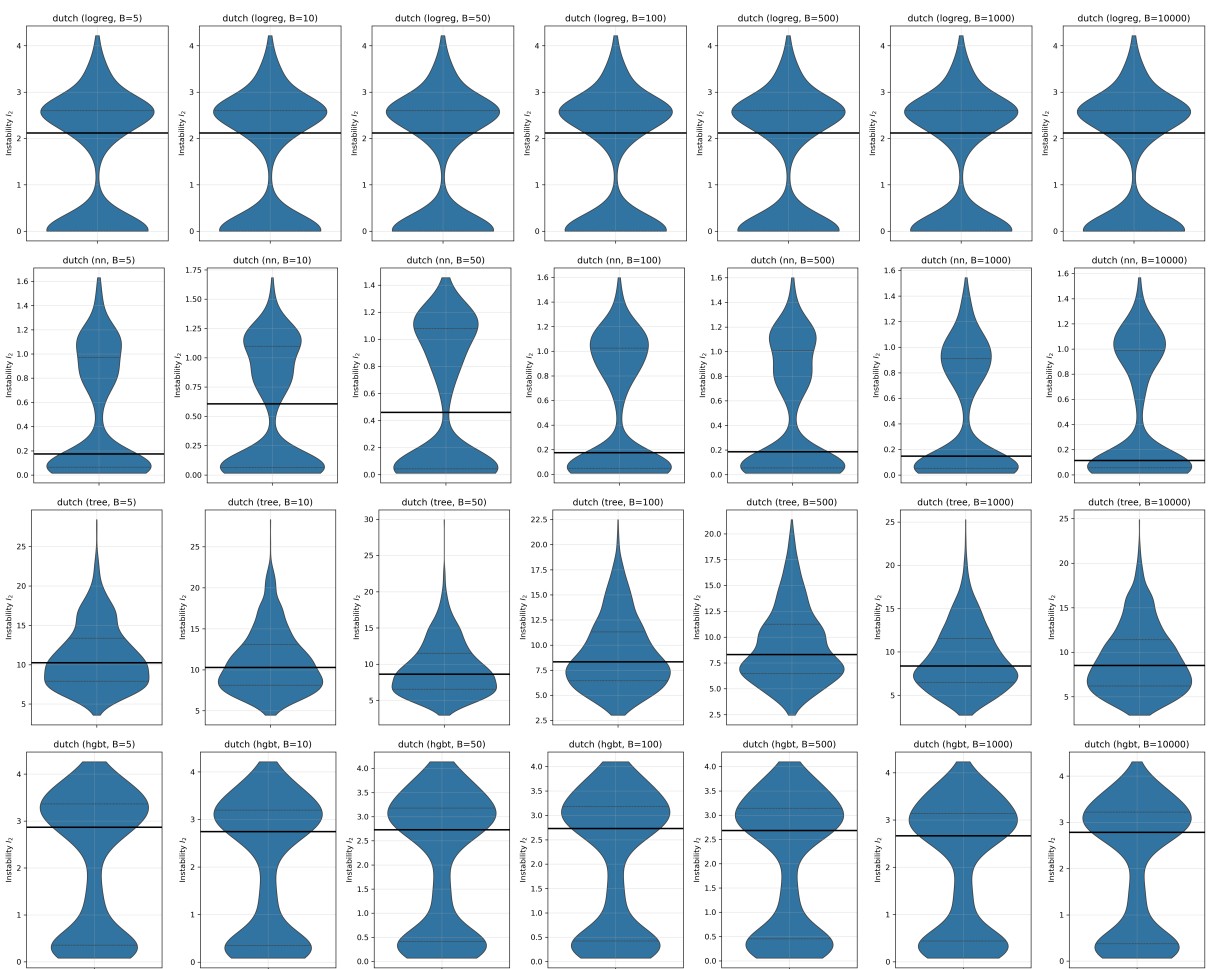

**Figure B.14.** Instability $I_2$ distributions per model and $B$ on `Dutch`.

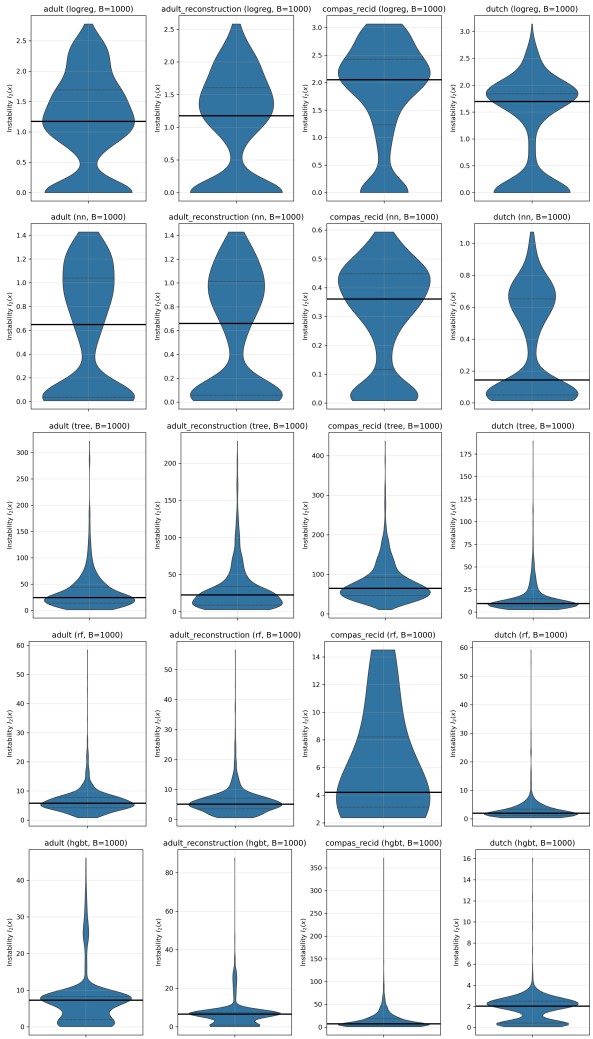

**Figure B.15.** Normalised Instability $I_2$ distributions per model and $B = 1000$.

