# OpenReview forum: "Assessing the Fragility of SHAP-Based Model Explanations Using Counterfactuals"
_NLDL.org/2026/Conference — NLDL 2026 Oral_

### Official Review · Reviewer_x1GQ · 2025-09-30
**Work that investigates procedural fairness of machine learning models in several tabular settings**

**Rating:** 4
**Confidence:** 3
**Final Rating:** 4
**Final Confidence:** 3

**Summary:**

The authors investigate procedural fairness, which examines whether a model's decisions remain stable in the face of small, plausible changes to an input. While the concept of using counterfactuals to test this is not new, the paper effectively argues that current methods for creating these test cases are flawed, often producing unrealistic examples through random noise or simple generative artifacts from a counterfactual generator.

To address this gap, the paper presents a model and task-agnostic methodology designed to generate more realistic counterfactuals. The process is multi-staged: first, they generate a diverse set of candidates using DiCE with multiple seeds. After deduplicating this pool, they select the most plausible counterfactual using a KNN algorithm guided by per-feature statistics. This ensures the final test case is a close neighbor in the data distribution, better reflecting real-world perturbations.

The authors applied this framework in an empirical study across four tabular datasets, using SHAP values to interpret the behavior of Linear Regression, MLP, and Decision Tree models. Their analysis yielded several findings. For instance, they demonstrated that model instability consistently increases as counterfactuals are generated closer to the decision boundary. Their results also show that Decision Trees exhibit significantly lower stability compared to both Linear Regression and MLPs. Furthermore, their analysis revealed that certain protected classes exhibited larger changes in the SHAP values under plausible edits.

**Strengths:**

- Proposes a framework for evaluating the procedural fairness of machine learning models that is agnostic to architecture and task.
- The counterexamples are designed to be as close to a selected instance through a combination of sampling multiple seeds, distance control based on per-feature distance, and being derived from a separate split in the dataset.
- The framework suggests that linear regression models and MLPs are generally more stable than their tree counterparts. This is useful information that will inform the creation of systems that need to deal with procedural stability.
- The authors are thorough in explaining all their hyperparameters, data pre-processing steps, and the components of the framework. Reproducibility should not be too difficult.

**Weaknesses:**

- In the field of uncertainty estimation, it is generally the case that ensembles are the most robust models. Furthermore, ensemble models like gradient boosting are typically state-of-the-art in most tabular settings. The paper acknowledges this; however, the authors do not make a comparison against such a collective (random forests, gradient boosted trees, or an ensemble of MLPs), which makes it difficult to predict the efficacy of the framework in real-world tasks.
- The method that the authors used to generate counterexamples using DiCE and KNN might still generate implausible examples despite being distance-controlled. I imagine usage in real-world scenarios might still require a more nuanced counterfactual generation that requires background knowledge.
- (minor) I found the approach a bit complex and difficult to parse. Simpler writing and an algorithm in the appendix would make it much easier for a reader to understand the whole framework.
- (minor) It appears that the authors argue that their approach is practical. It would be useful to see the overhead for assessing fragility in wall clock time to provide empirical evidence for this claim.

**Final Justification:**

The authors have successfully remedied my critiques:
- Will provide ensemble models such as random forests and gradient boosted trees in the camera-ready version of the paper
- Will provide a breakdown of wall-clock time for the entire process as well
- Fixed the typos and writing issues raised by other reviewers, and also added an algorithm to the appendix

Based on the new information, I maintain my recommendation of accept. While I appreciate the planned additions and improvements, a substantial portion of the new material—particularly the ensemble results—is not yet available on OpenReview. This makes it difficult to fully assess the quality and contribution of the forthcoming updates to the community.

**Justification:**

I find the core idea compelling. However, the lack of tests conducted on ensembles does not judge the efficacy of the proposed framework using state-of-the-art tabular methods. There is also the problem of the counterfactuals potentially being unrealistic, and it is not clear from the text how the counterfactual generation could be modified to account for this.

---

> ### Author Rebuttal · Authors · 2025-10-22
>
> We thank the reviewer for their careful reading and their valuable comments.
>
>
> > In the field of uncertainty estimation, it is generally the case that ensembles are the most robust models. Furthermore, ensemble models like gradient boosting are typically state-of-the-art in most tabular settings. The paper acknowledges this; however, the authors do not make a comparison against such a collective (random forests, gradient boosted trees, or an ensemble of MLPs), which makes it difficult to predict the efficacy of the framework in real-world tasks.
>
> Thank you for pointing this out. We agree that adding more models (and metrics) would make our paper stronger. We thus added a random forest classifier and a gradient boosting classifier to our experiments. We then re-ran the entire audit under the same SHAP configuration and the same counterfactual pipeline. In addition, we now also report F1 and AUC metrics.
>
>
> > The method that the authors used to generate counterexamples using DiCE and KNN might still generate implausible examples despite being distance-controlled. I imagine usage in real-world scenarios might still require a more nuanced counterfactual generation that requires background knowledge.
>
> In our paper, we use widely adopted and public fairness benchmarks and keep the audit fully automated to ensure reproducibility and ease of use. We believe that deployment in real-world settings usually necessitates domain knowledge (e.g. immutability, legal/actionability rules and realistic ranges). Thus, evaluating our approach in a real-world scenario is non-trivial and we believe it is out of scope for this submission, but valuable future work.
>
> Our protocol is generator-agnostic and compatible with such additions; practitioners can plug in stricter CF generators or add plausibility filters without altering our seed pooling, $K$-nearest neighbor selection or SHAP shift summaries. We now explicitly state this and have expanded the Limitations section accordingly.
>
>
> > (minor) I found the approach a bit complex and difficult to parse. Simpler writing and an algorithm in the appendix would make it much easier for a reader to understand the whole framework.
>
> This is a very valid and important point. We acknowledge that our approach might be hard to understand and we are committed to improve its readability. We added an algorithm and adapted some sentences in our paper and hope that those changes increase the readability of our approach.
>
>
> > (minor) It appears that the authors argue that their approach is practical. It would be useful to see the overhead for assessing fragility in wall clock time to provide empirical evidence for this claim.
>
> By "practical", we mean plug-and-play functionality across models and datasets, with one SHAP setup (a shared independent mask on logits), no retraining and batched inference using standard libraries. To accommodate your concern, we aim to report wall-clock times per dataset–model, split into DiCE vs. SHAP, in the appendix. In our evaluation, DiCE dominates runtime, while permutation SHAP with a fixed background scales roughly linearly with the number of evaluations.
>
> Our aim is to create a reproducible and model-agnostic diagnostic tool. Thus, our focus is on the audit rather than being time-efficient. Note that since our audit is generator-agnostic, faster CF solvers can be swapped in without changing the pipeline and metrics.
>
>
> > I find the core idea compelling. However, the lack of tests conducted on ensembles does not judge the efficacy of the proposed framework using state-of-the-art tabular methods. There is also the problem of the counterfactuals potentially being unrealistic, and it is not clear from the text how the counterfactual generation could be modified to account for this.
>
> Thank you again for your feedback and your positive assessment of our work! Your feedback helped us to improve our paper. We hope that we were able to address all of your remaining concerns.

---

### Official Review · Reviewer_Vg63 · 2025-10-03
**Assessing Explanation Fragility of SHAP using Counterfactuals**

**Rating:** 2
**Confidence:** 3
**Final Rating:** 4
**Final Confidence:** 3

**Summary:**

This paper proposes a method to evaluate the stability of feature importance obtained by SHAP. Using the log-odds (logit) as the basis, the authors assess stability across three models: logistic regression, decision tree, and neural network. Specifically, they examine how feature importances change when counterfactuals are generated using DiCE. They further evaluate combinations of these three models with several datasets. Based on numerical experiments with practical data, the paper reports how stability depends on model type and also investigates how stability varies with the distance from the decision boundary.

**Strengths:**

The investigation of model-dependent stability is particularly interesting. The authors show that decision-tree models yield less stable importances, whereas neural networks are comparatively more stable. They also highlight the connection to fairness, which is another important and timely topic. In addition, the paper carefully addresses the potential confounding effects of random seeds by aggregating results across multiple seed values.

**Weaknesses:**

First, I am concerned that the definition of $K_{\mathrm{eff}}(x)$ on p.3 does not work well, since it always evaluates to be positive as long as the $K$-nearest neighbor rule is applied. Because no threshold is introduced at this stage, $S_K(x)$ will always be non-empty. Indeed, the authors acknowledge at the beginning of Section~5 that in numerical experiments this value was always positive. Consequently, the method as currently defined does not seem able to control the effect of the distance from the decision boundary, which casts doubt on the method’s validity and leads me to a negative assessment

Second, since the instability of feature importance is often measured via the Lipschitz coefficient of the model, it would be more appropriate to normalize the difference by distance. In other words, in the definition of $\widetilde{\Delta}_j(x)$ in Section~5, I suggest dividing $|\phi_j(x)-\phi_j(z)|$ by $\|x-z\|$.

Third, the dependency on the model may vary with the extent of generalization. Therefore, it would be helpful for the authors to clarify whether feature importance was measured on training or test data, and to report additional performance metrics (e.g., F1-score, AUC) alongside accuracy.

Finally, there are several typos and inconsistencies throughout the manuscript, which should be corrected. For example:

 p.1, l.35: ``small, small, feasible'' $\to$ redundant repetition

p.4, l.376: ``bins what preserve the horizontal axis''

$\to$   ``bins that preserve the horizontal axis''


p.5, l.450: ``and explaining SHAP on logits yields''

 $\to$ ``yield''   (Moreover, $L_f$ is not defined anywhere.)


p.5, l.456: ``Thresholds use $\hat{p}(x)>0.5$''

 $\to$ sentence appears incomplete or unclear


p.7, l.585: extraneous ``3'' before a comma

p.7, l.618: ``instability see in Figure B.4.''

 $\to$ ``instability is seen in Figure B.4.''

p.7, l.663: ``proximity bins did group 1'' $\to$ awkward phrasing, consider revision

p.8, l.669: discrepancy between text ($\delta=0.0028$) and Table~2 ($\delta=0.028$)

**Final Justification:**

I recognize the authors' efforts to clarify my earlier misunderstanding and their willingness to revise the paper accordingly. Moreover, I acknowledge that they have added additional metrics and re-conducted experiments to strengthen their results. Based on these, personally I'd like to change (improve) my rating.

**Justification:**

Although the observation that instability depends on the model is interesting, I believe it cannot be substantiated without additional indicators that reflect the extent of generalization, such as AUC, F1-score, or balanced accuracy. Moreover, the definition of “coverage” appears problematic: since $K_{\mathrm{eff}}(x)$ is always positive, the claimed effect of distance control seems doubtful. Instability would be more appropriately quantified via the Lipschitz coefficient, rather than merely the raw difference in feature importance.

In summary, although the paper presents an appealing direction, I find that there remains considerable room for improvement before it can be considered ready for publication.

---

> ### Author Rebuttal · Authors · 2025-10-22
>
> We thank the reviewer for their careful reading and their valuable comments.
>
> > First, I am concerned that the definition of $K_\text{eff}(x)$ on p.3 does not work well, since it always evaluates to be positive as long as the $K$-nearest neighbor rule is applied. Because no threshold is introduced at this stage, $S_K(x)$ will always be non-empty.
> > Indeed, the authors acknowledge at the beginning of Section~5 that in numerical experiments this value was always positive.
> > Consequently, the method as currently defined does not seem able to control the effect of the distance from the decision boundary, which casts doubt on the method’s validity and leads me to a negative assessment.
>
> There might be a misunderstanding. Our selection step does not guarantee $K_\text{eff}(x)>0$!
>
> 1. We first built a finite pool of feasible, decision-flipping counterfactuals by pooling an deduplication DiCE outputs.
>
> 2. We then take the $K$ nearest $S_K(x)$ from that pool, called $K_\text{eff}(x) = \min\\{K, |\tilde{C}(x)|\\}$.
>
> If there is no feasible counterfactual found, then $\tilde{\mathcal{C}}(x)=\emptyset$ and $|S_K(x)|=0$, therefore $K_\text{eff}(x)=0$. We will clarify this in the paper.
>
> Note that the distance is controlled in two ways:
>
> 1. the selection metric $D(u,v) = ||T(u)-T(v)||_2$ uses standardized features, ensuring comparability across coordinates; and
>
> 2. we condition at analysis time on proximity to the decision boundary using equal-width bins of $|\ell(x)|$, which isolates boundary effects when comparing instability across groups.
>
> This analysis-time control avoids feasibility-induced selection bias from a hard radius.
>
>
> > Second, since the instability of feature importance is often measured via the Lipschitz coefficient of the model, it would be more appropriate to normalize the difference by distance. In other words, in the definition of $\tilde\Delta_j(x)$ in Section~5, I suggest dividing $|\phi_j(x)-\phi_j(z)|$ by $|x-z|$.
>
> We appreciate your suggestion. We now determined the distance-normalized instability and will report it alongside with our raw metric to accommodate your concern.
>
> However, we will keep the raw metric as the primary diagnostic and use the normalized metric as a robustness check, since we first want a fairness risk signal and if a group must move farther to cross the boundary, $I_2$ captures the absolute change in reasons a person actually experiences under the required edit. Normalizing by distance can hide this burden. Moreover, in Fig. 2, we analyze instability as a function of proximity. Normalizing introduces a varying denominator across bins, so trends across bins are not directly comparable.
>
>
> > Third, the dependency on the model may vary with the extent of generalization. Therefore, it would be helpful for the authors to clarify whether feature importance was measured on training or test data, and to report additional performance metrics (e.g., F1-score, AUC) alongside accuracy.
>
> Thank you for this very valid suggestion. The SHAP-based feature importance is computed on the test-set. The same holds for the other metrics (e.g. accuracy). We agree that additional metrics would be beneficial. Thus, we re-ran our experiments, computed these additional metrics, and added them to our paper.
>
>
> > Finally, there are several typos and inconsistencies throughout the manuscript, which should be corrected.
>
> Thank you for pointing out those problems. We updated our paper.
>
>
> > In summary, although the paper presents an appealing direction, I find that there remains considerable room for improvement before it can be considered ready for publication.
>
> Thank you again for your valuable feedback; it helped to improve our paper! We hope that we were able to address your remaining concerns and hope that you reconsider your assessment of our work.

---

### Official Review · Reviewer_bB97 · 2025-10-08
**A disciplined diagnostic that audits SHAP (in)stability using plausible counterfactuals**

**Rating:** 5
**Confidence:** 4
**Final Rating:** 5
**Final Confidence:** 4

**Summary:**

The paper audits SHAP stability using plausible counterfactuals.
They pool DiCE counterfactuals across seeds, deduplicate the candidates, retain the $K$-nearest in standardized feature space, and then measure per-feature SHAP attribution shifts between each instance and its counterfactuals.
Instance-level instability is summarized by $L_1$ and $L_2$ norms of the per-feature median absolute attribution changes.
The main finding is that instability increases near the decision boundary. A decision tree exhibits higher typical and tail instability than LR/MLP. After proximity control, most model-dataset pairs overlap across groups, with a few small yet consistent residual gaps that the authors interpret as procedural fairness risk signals.
Main findings is that instability rises with boundary distance across models and datasets. Trees are uniformly higher. Actionable risk signals appear only in specific dataset-model pairs (e.g., Compas-NN) after proximity matching.

**Strengths:**

The paper presents a clear and reproducible audit protocol that reduces counterfactual generator variance through seed pooling, deduplication, and distance control. It employs a unified SHAP configuration across models, using a shared background and logit space to enhance comparability and internal validity. The inclusion of diagnostic indicators helps disentangle generator artifacts from genuine model behavior. The proximity matched group analysis, supported by bootstrap CIs and Cliff’s delta, provides robust group comparisons. The work is appropriately positioned as diagnostic rather than normative, avoiding overclaiming and encouraging complementary use alongside accuracy and outcome-based fairness evaluations.

**Weaknesses:**

The paper’s conceptual novelty is limited, as it mainly systematizes existing ideas on explanation (in)stability rather than introducing a new theoretical perspective.
Its reliance on interventional SHAP with an independent masker may distort attributions when features are correlated, but the authors acknowledge this limitation in the manuscript and discuss its potential effects on stability estimation.
The scope remains restricted to tabular data and SHAP-based explanations, limiting generalizability to other modalities or explanation families.
The paper would further benefit from a lightweight theoretical analysis that formally links (in)stability measures to model smoothness and counterfactual distance, providing a clearer analytical foundation for the observed empirical trends.

Minor suggestions
- Since DiCE may be less familiar to some readers than SHAP, consider introducing its full name (e.g., Diverse Counterfactual Explanations) at first mention, for example around L10.
- At L80, replacing the period with a comma in “from model behavior,” improves grammatical flow and preserves the sentence’s continuity.
At L669, the reported delta value appears inconsistent: the text lists 0.0028, but the corresponding table indicates 0.028. Please verify and correct the number for accuracy.

**Final Justification:**

Although there is room for further improvement in the current manuscript, Reviewer bB97 is overall satisfied with the authors’ rebuttal and the interaction to the other reviewers. The reviewer looks forward to the revised manuscript that addresses the comments as promised if this paper is accepted.

**Justification:**

The paper delivers a careful and useful diagnostic for explanation (in)stability. The protocol is disciplined and the findings are well contextualized. Although the conceptual step is incremental, the work has clear practical value for reliability audits of post-hoc explanations. With the suggested ablations and minor writing fixes, it will serve as a solid reference.

---

> ### Author Rebuttal · Authors · 2025-10-22
>
> We thank the reviewer for their careful reading and their valuable comments.
>
> > The paper’s conceptual novelty is limited, as it mainly systematizes existing ideas on explanation (in)stability rather than introducing a new theoretical perspective. Its reliance on interventional SHAP with an independent masker may distort attributions when features are correlated, but the authors acknowledge this limitation in the manuscript and discuss its potential effects on stability estimation.
> > The scope remains restricted to tabular data and SHAP-based explanations, limiting generalizability to other modalities or explanation families.
> > The paper would further benefit from a lightweight theoretical analysis that formally links (in)stability measures to model smoothness and counterfactual distance, providing a clearer analytical foundation for the observed empirical trends.
>
> We mainly agree with you. A generalization of our approach beyond tabular data as well as a theoretical analysis would both be worth having. However, we believe that it is out of scope for this paper as we already reach the page limit and we see it as future work.
>
>
> > Since DiCE may be less familiar to some readers than SHAP, consider introducing its full name (e.g., Diverse Counterfactual Explanations) at first mention, for example around L10.
> > At L80, replacing the period with a comma in “from model behavior,” improves grammatical flow and preserves the sentence’s continuity. At L669, the reported delta value appears inconsistent: the text lists 0.0028, but the corresponding table indicates 0.028. Please verify and correct the number for accuracy.
>
> Thank you for pointing out those issues. We revised our paper to increase its readability. Furthermore, we now also report F1 and AUC metrics and added an algorithmic description of our approach.
>
>
> > The paper delivers a careful and useful diagnostic for explanation (in)stability. The protocol is disciplined and the findings are well contextualized. Although the conceptual step is incremental, the work has clear practical value for reliability audits of post-hoc explanations. With the suggested ablations and minor writing fixes, it will serve as a solid reference.
>
> Thank you again for your feedback and your very positive assessment of our work! Your feedback helped us to improve our paper.

---

### Official Review · Reviewer_UFqr · 2025-10-11
**review of the submitted paper**

**Rating:** 4
**Confidence:** 4

**Summary:**

The authors devise a protocol to measure stability of attributions for models over tabular data where the attributions are computed using SHAP. They perform experiments using 4 datasets and 3 models. To this end they generate counterfactuals and compute median deviations of attributions between the original sample and the counterfactual. They accumulate the deviations over features and over samples. They compare their devised measures for certain groups in fairness datasets.

**Strengths:**

- A non-trivial idea.
- They explain the used statistical measures in detail.
- Violin plots are a good choice to show the median differences on population level.
- Discussion of limitations.

**Weaknesses:**

- probably the largest weakness is the selection of B:

To make explanations comparable 475
within a dataset we fix a single B and reuse it across 476
all models for that dataset. The background acts as 477
a Monte Carlo sample for the interventional expec- 478
tation.
p Its sampling error decreases on the order of 479
O(1/ |B|) while runtime and memory grow with 480
|B|. We form B by label- and group-stratified ran- 481
dom subsampling of the training inputs with a fixed 482
seed and a fixed budget |B| = 300. This keeps 483
B in-sample, represents both outcome classes and 484
sensitive-attribute groups, and removes background- 485
induced variation by reusing the same B.

  - While this is in-sample, it might be not a real background in the sense of absence of information. In particular, if the protected group is a small subset, the B might not be representative for it and the values of B might be far away from it, thus causing a larger SHAP instability for the protected group .

- Some things are hard to read.

"but as a probe of plausible, policy-compliant variation:"

are a model’s local ex-
planations stable under small, policy-compliant edits
that flip the decision

- what is the meaning of policy-compliant here ?

Some unclarities:

- It is not clear what "We then select the K nearest counterfactuals" in lines 283 284 should mean. If they mean the top-k closest counterfactuals to the original sample x, then K_eff(x) should be trivially = K and coverage should be 1.
- procedural uncertainty risk needs an explanation

- One would think that it is not an assessment of SHAP as suggested by the title but rather of the model under SHAP explanations ...



minor:
- \Delta_{med} in line 331 and \Delta^{(g,g')} in line 352 seems to be the same ?
- typos: small, small, feasible
perturbations that cross the decision boundary

**Justification:**

While the paper has some unclarities, it is overall worth reading and worth to be distributed within the community.

---

> ### Author Rebuttal · Authors · 2025-10-22
>
> We thank the reviewer for their careful reading and their valuable comments.
>
> > probably the largest weakness is the selection of B:
> >
> > *(quotes lines 475-486)*
> >
> > While this is in-sample, it might be not a real background in the sense of absence of information. In particular, if the protected group is a small subset, the B might not be representative for it and the values of B might be far away from it, thus causing a larger SHAP instability for the protected group .
>
> Note that we use interventional SHAP with an independent mask and a fixed background $B$ drawn from the training split. This serves as a reference distribution for imputations and is chosen in-sample (label- and group-stratified). The same $B$ is then reused across models to ensure comparability within the dataset.
>
> We understand your concern. Thus, we re-ran our experiments and produced Fig. 1 for the following choices: $B \in \\{100, 300, 1000, 10000, n\\}$, where $n$ is the training-set size, keeping the same counterfactuals and only changing $B$. Overall, we observe no drastic differences; the strongest difference is in the tree model. Besides, the distance-instability trend and our proximity-matched group findings remain qualitatively stable.
>
> To address your concern, we update all numbers and figures in our paper and use $B=1000$ as a default. Furthermore, we will discuss the choice of $B$ in the appendix and show the effect of $B$ on on representative models/datasets.
>
>
> > Some things are hard to read.
> >
> > "but as a probe of plausible, policy-compliant variation:"
> >
> > are a model’s local explanations stable under small, policy-compliant edits that flip the decision
> >
> > what is the meaning of policy-compliant here ?
>
> By "policy-compliant" we simply mean feasible within the counterfactual generator's search space (DiCE), i.e., candidates that respect variable types and stay within data-derived bounds. However, we do not freeze any attributes in our experiments. To avoid confusion, we will replace "policy-compliant" with: "feasible within DiCE's search space: valid encodings and data-range bounds." We will also add one sentence in Methods clarifying the feasibility setting used per dataset.
>
>
> > It is not clear what "We then select the $K$ nearest counterfactuals" in lines 283 284 should mean. If they mean the top-$k$ closest counterfactuals to the original sample $x$, then $K_\text{eff}(x)$ should be trivially = $K$ and coverage should be 1.
>
> Not exactly, as $K_\text{eff}(x) = \min\\{K, |\tilde{C}(x)|\\}$ because KNN operates on the deduplicated, finite candidate pool of counterfactuals for $x$, called $\tilde{C}(x)$. When the generator yields fewer than $K$ unique candidates (or none), then $K_\text{eff}(x) \leq K$. We will clarify this in the paper.
>
>
> > procedural uncertainty risk needs an explanation
>
> We use the term "procedural-fairness risk signal" to refer to a diagnostic indication that, under our fixed protocol, similar individuals receive less stable explanations. This is a concern about the process (the reasons given) rather than the outcomes themselves. It is not a normative fairness claim.
>
>
> > One would think that it is not an assessment of SHAP as suggested by the title but rather of the model under SHAP explanations ...
>
> Thank you for pointing this out. We agree that our target is the model-explainer system, not SHAP in isolation. We have now clarified this in the abstract, introduction, methods, and limitations sections. Our metric evaluates the stability of SHAP explanations for a fixed trained model and shared background. We will also try to adapt the title.
>
>
> > $\Delta_\text{med}$ in line 331 and $\Delta^{(g,g')}$ in line 352 seems to be the same ?
> >
> > typos: small, small, feasible perturbations that cross the decision boundary
>
> Thank you for pointing out those problems. You are right and we updated our paper.
>
>
> > While the paper has some unclarities, it is overall worth reading and worth to be distributed within the community.
>
> Thank you again for your feedback and your positive assessment of our work! Your feedback helped us to improve our paper. We hope that we were able to address all of your remaining concerns.

---

### Meta-Review · Area_Chair_kizZ · 2025-10-31

**Recommendation:** Accept (Oral)
**Confidence:** 4

**Metareview:**

After rebuttal, the reviewers all shared positive views of the paper. While the text can be further improved for clarity (incorporating some of the author's suggestions in the response), the work offers an interesting tool for assessing SHAP (in)stability. The authors’ careful responses and engagement with the process have addressed all the initial concerns and demonstrate a clear understanding of the limitations of the approach. As is, it is a valuable contribution on a timely topic to be further discussed within the community.

---

### Decision · Program_Chairs · 2025-11-05

**Decision:**

Accept (Oral)

**Comment:**

We recommend an oral and a poster presentation given the AC and reviewers recommendations.